# PI3Kγ stimulates a high molecular weight form of myosin light chain kinase to promote myeloid cell adhesion and tumor inflammation

Michael C. Schmid[1✉], Sang Won Kang[2], Hui Chen[1], Marc Paradise[1], Anghesom Ghebremedhin[1], Megan M. Kaneda[1], Shao-Ming Chin[1], Anh Do[1], D. Martin Watterson[3] & Judith A. Varner [1,4✉]

Myeloid cells play key roles in cancer immune suppression and tumor progression. In response to tumor derived factors, circulating monocytes and granulocytes extravasate into the tumor parenchyma where they stimulate angiogenesis, immune suppression and tumor progression. Chemokines, cytokines and interleukins stimulate PI3Kγ-mediated Rap1 activation, leading to conformational changes in integrin α4β1 that promote myeloid cell extravasation and tumor inflammation Here we show that PI3Kγ activates a high molecular weight form of myosin light chain kinase, MLCK210, that promotes myosin-dependent Rap1 GTP loading, leading to integrin α4β1 activation. Genetic or pharmacological inhibition of MLCK210 suppresses integrin α4β1 activation, as well as tumor inflammation and progression. These results demonstrate a critical role for myeloid cell MLCK210 in tumor inflammation and serve as basis for the development of alternative approaches to develop immune oncology therapeutics.

[1] Moores Cancer Center, University of California, San Diego, La Jolla, CA, USA. [2] Department of Life Science, Ewha Womans University, Seoul 03760, Republic of Korea. [3] Department of Pharmacology, Northwestern University, Chicago, IL, USA. [4] Department of Pathology and Medicine, University of California, San Diego, La Jolla, CA, USA. ✉email: M.Schmid@liverpool.ac.uk; jvarner@health.ucsd.edu

Inflammation contributes to infectious, cardiovascular, auto-immune, and neurodegenerative diseases and cancer[1]. Vascular endothelial cells and myeloid cells (monocytes, macrophages, and neutrophils) play critical roles in inflammatory responses. Myeloid cells invade inflamed tissues, where they can contribute to pathology by expressing inflammatory mediators, reactive oxygen species, angiogenic factors, and immunosuppressive factors[2]. Endothelial cells respond to inflammatory stimuli by promoting myeloid cell adhesion and extravasation, by becoming more permeable and by undergoing angiogenesis[3]. Sustained recruitment of inflammatory cells to tissues can lead to progressive damage induced by release of oxygen radicals, which can lead to the development of cancer[4]. Targeting the mechanisms controlling myeloid cell responses to inflammatory stimuli could lead to additional approaches to suppress chronic inflammation and cancer.

A variety of inflammatory factors released by damaged tissues activate G protein coupled receptors (GPCRs), receptor tyrosine kinases (RTKs) or Toll-like receptor/interleukin1 receptor family members (TLR/IL1Rs) to initiate myeloid cell recruitment during inflammation[5]. In previous studies, we demonstrated that myeloid cell recruitment during tumor inflammation depends on the VCAM-1 receptor integrin α4β1 but not on other integrins[6–8]. Inactivation of or deletion of integrin α4β1 in myeloid cells prevented monocyte and granulocyte extravasation and suppressed tumor inflammation and growth, as did selective antibody and small molecule antagonists of this integrin[6,7]. We found that inflammatory stimuli promote integrin α4β1 activation, cell adhesion, and myeloid cell trafficking to tumors by activating a single PI3K isoform, PI3Kγ[7]. PI3Kγ mediated integrin activation requires PLCγ, which cleaves phospholipids to produce Ca2+ and diacylglycerol (DAG), the DAG-activated RapGEFs, CalDAG-GEFI and II, Rap1a and RIAM, thereby activating integrin α4β1 in a RIAM-dependent manner[7,9]. Pharmacological or genetic blockade of PI3Kγ or integrin α4β1 suppressed adhesion and recruitment of both monocytes and granulocytes into tumors and inhibited disease progression[6,7,9]. Our studies and those of others have shown that integrin α4β1 conformational changes (activation) depend on Rap1 (Ras-proximate-1), a Ras-like small GTP-binding protein and its effector protein RIAM, which localizes talin as well as paxillin to the integrin cytoplasmic tails[9–13]. While talin binds integrin β chain cytoplasmic tails, inducing a shift in the conformation of the extracellular domain of the integrin and resulting in an allosteric increase in ligand-binding affinity[13], paxillin binds to the integrin α4 cytoplasmic tail, promoting α4 integrin conformation changes that promote adhesion and trafficking of lymphocytes and myeloid cells[6,9–12]. These findings indicate that targeting the signaling events that lead to inflammation might provide significant benefit in the treatment in cancer. Deciphering the molecular mechanisms by which myeloid cell integrins are activated could lead to the development of approaches to treat acute and chronic inflammatory and vascular diseases.

Here we show that PI3Kγ activates a high molecular weight form of myosin light chain kinase, MLCK210, leading to myosin-dependent activation of Rap1 GTP loading and integrin α4β1 conformational changes that promote tumor inflammation and progression. These studies also identify MLCK210 inhibitors as potential immune oncology therapeutics.

## Results

**MLCK210 colocalizes with integrin α4β1.** We previously determined that tumor derived growth factors and chemokines, such as SDF1α (CXCL12), TGFβ, and IL1β, promote PI3Kγ-mediated integrin α4β1 conformational changes and adhesion to

endothelium in circulating myeloid cells during tumor inflammation[7,9]. To identify proteins that closely associate with myeloid cell integrin α4β1 that might regulate its activation, we stimulated primary bone-marrow-derived CD11b+ myeloid cells with Lewis Lung carcinoma tumor cell condition medium (TCM) or basal media, then solubilized cells and immunoprecipitated integrin α4β1. We then performed silver staining and immunoblotting to detect integrin α4β1 interacting proteins (Fig. 1a). Immunoblotting revealed that paxillin and talin co-immunoprecipitated with integrin α4β1 upon stimulation with TCM (Fig. 1a), as predicted based upon our prior published studies[9–12]. Silver staining identified two major co-precipitating proteins with molecular weights of 80 kDa and 210 kDa upon stimulation with TCM but not under basal conditions (Fig. 1a; Supplementary Fig. 1a). These proteins were excised from silver-stained gels and subjected to protein sequencing by tandem mass spectrometry (Fig. 1b). Tandem mass spectrometry analysis identified the 80 kDa co-precipitating protein as Gelsolin and the 210 kDa co-precipitating protein as a high molecular weight form of myosin light chain kinase (MLCK) (Fig. 1b; Supplementary Fig. 9). A 210 kDa isoform of MLCK was previously shown to play a key role in regulating vascular permeability[14–19]. Gelsolin is an actin filament severing and capping protein in podosomes that is inhibited by PIP3, the product of PI3Kinases[20]. MLCK210 is an alternatively spliced MLCK isoform that contains the entire 108 kDa non-muscle MLCK protein as well as six N-terminal IgG-C2-like domains and two tandem DXR actin-binding domains (Supplemental Fig. 1b)[14–18]. MLCKs contain a core catalytic domain and a calmodulin-sensitive auto-inhibitory domain that folds back on the catalytic domain to inhibit kinase activity. Upon activation by calmodulin, MLCK phosphorylates Myosin Light Chain (MLC), thereby stimulating myosin ATPase[21]. As activated myosin undergoes a conformational change that allows it to cyclically bind and release actin filaments, it promotes contractility that leads to cell shape change and migration[22].

To determine whether MLCK210 and integrin α4β1 colocalize in intact plasma membranes, CD11b+ myeloid cells from wildtype (WT) and mutant mice in which the alternatively transcribed isoform of *Mylk* encoding MLCK210 was selectively deleted without impacting MLCK108[15] (here designated *Mlck210*−/− mice), were incubated with basal culture medium or culture medium containing the chemokine SDF1α. Plasma membranes were purified, solubilized, and immunoblotted to detect integrin α4 and MLCK210. MLCK210 was detected in plasma membranes only from SDF-1α-stimulated wildtype (WT) cells but not in from *Mlck210*−/− cells or basal WT cells (Fig. 1c). In contrast, integrin α4 was constitutively associated with plasma membranes (Fig. 1c). These results indicate that MLCK210 associates with the membrane only upon chemokine receptor signaling while integrin α4β1 is constitutively present in plasma membranes.

To examine the role of MLCK210 in the distribution of integrin α4 in myeloid cells, myeloid cells were incubated with SDF-1α or BSA coated 0.9 μm microbeads, fixed, permeabilized, and analyzed for clustering or aggregation of integrin α4 (green) by immunofluorescence microscopy. We found that SDF-1α but not BSA induced clustering or aggregation of integrin α4 in WT but not *Mlck210*−/− myeloid cells (Fig. 1d–e). Additionally, myeloid cells were transfected with control All Stars non-silencing or *Mylk* siRNA, then incubated with SDF-1α-coated beads. Western blotting verified MLCK210 gene knockdown (Fig. 1f). Confocal microscopy assessment of the effect of gene knockdown on integrin localization showed that SDF-1α beads induced integrin clustering in All Stars non-silencing siRNA-transfected cells but not in *Mylk* siRNA-transfected cells (Fig. 1g).

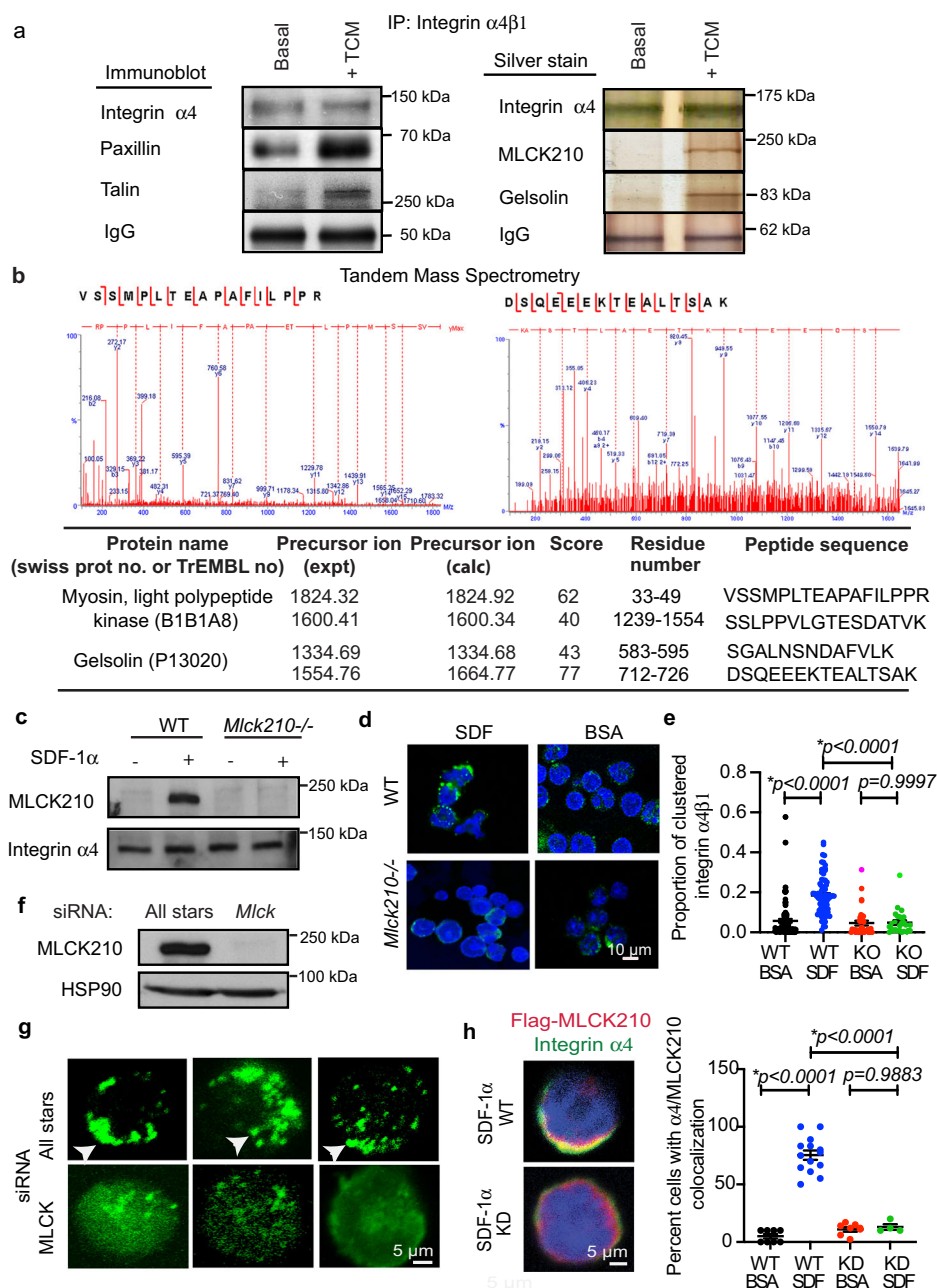

To determine whether MLCK210 and integrin α4β1 colocalize in the same membrane domains upon chemokine stimulation, we transiently transfected *Mlck210*$^{-/-}$ cells with Flag-tagged wild-type and kinase-dead MLCK210. Because antibodies directed against MLCK react with multiple MLCK isoforms, selective detection of MLCK210 by immunofluorescence in intact cells could only be accomplished by transfecting *Mlck210*$^{-/-}$ cells with these Flag-tagged constructs. Flag-tagged wildtype and kinase-dead *Mlck210* transfected myeloid cells were incubated with SDF-1α or BSA coated 0.9 μm microbeads, fixed, permeabilized, and analyzed for colocalization of MLCK210 (red) and integrin α4 (green) by immunofluorescence microscopy. We found that SDF-1α but not BSA induced colocalization of MLCK210 (red) and integrin α4 (green) in *Mlck210* WT but not KD transfected cells (Fig. 1h; Supplementary Fig. 2a–c). MLCK210 and integrin α4 overlap in the merged images can be observed as regions of yellow color indicated by arrows. Additionally, we found that SDF-1α but not BSA induced colocalization of paxillin (red) and integrin α4 (green) in *Mlck210* WT but not KD transfected cells (Supplementary Fig. 2d, e). Taken together, these results demonstrate that MLCK210 closely associates with integrin α4 in the plasma membrane and is required to promote integrin α4β1 clustering in response to chemokine stimulation.

**MLCK210 is required for integrin α4β1-mediated cell adhesion.** Our prior studies showed that integrin α4β1 mediates myeloid cell adhesion to vascular cell adhesion molecule 1 (VCAM-1) on endothelium, leading to tumor inflammation and growth[6–8]. We found that a wide array of chemokines and cytokines stimulated adhesion to endothelial monolayers (HUVEC) or VCAM-1-coated substrates of CFDA-labeled peripheral blood CD11b+ myeloid cells from WT mice but not from mice with an integrin α4Y991 inactivating mutation (Fig. 2a). To

**Fig. 1 MLCK210 is an integrin α4β1-associated protein. a** Integrin α4β1 immunoprecipitates from tumor conditioned medium (TCM) stimulated or unstimulated (Basal) CD11b+ myeloid cells were silver-stained or immunoblotted to detect co-precipitating proteins. (Left) Integrin α4, paxillin, talin, and IgG were detected in α4 integrin immunoprecipitates by immunoblotting. Right: Co-immunoprecipitated integrin α4, 210 kDa (MLCK210) and 80 kDa (Gelsolin) proteins as well as a 50 kDa protein (IgG) were detected in α4 integrin immunoprecipitates by silver staining. These samples derived from the same experiments and silver-stained gels and immunoblots were processed in parallel. **b** Spectra and peptide sequences from tandem mass spectroscopy analysis of 80 kDa and 210 kDa proteins co-immunoprecipitated with integrin α4β1; these proteins were identified as gelsolin and a high molecular weight isoform of myosin light chain kinase, MLCK210, respectively. **c** Plasma membranes from unstimulated and SDF-1α stimulated WT and *Mlck210*[−/−] myeloid cells immunoblotted to detect MLCK210 and integrin α4. **d** Representative immunofluorescence images of integrin α4 (green) in BSA or SDF-1α stimulated WT and *Mlck210*[−/−] myeloid cells. Scale bars = 10 μm. **e** Proportion of clustered integrin α4β1 in individual WT cells stimulated with BSA (n = 126 cells) or SDF-1α (n = 73 cells) and in individual *Mlck210*[−/−] myeloid cells stimulated with BSA (n = 36 cells) or SDF-1α (n = 28 cells). p = 0.9247 WT BSA vs *Mlck210*[−/−] BSA, p < 0.0001 WT SDF-1α vs *Mlck210*[−/−] SDF-1α. Replicates are biologically independent samples. Statistical significance determined by one-sided Anova with Tukey's multiple comparisons. **f** Detection of MLCK210 and loading control HSP90 in immunoblots of lysates from *Mylk* (*Mlck*) and All Stars control siRNA-transfected myeloid cells. **g** Images of integrin α4 (green, arrows) in SDF-1α stimulated *Mylk* (*Mlck*) and non-silencing siRNA-transfected myeloid cells. **h** Images of SDF-1α stimulated *Mlck210*[−/−] myeloid cells transfected with wildtype (WT) or kinase dead (KD) Flag-tagged MLCK210 immunostained to detect integrin α4 (green) and MLCK210 (red). Nuclei were detected with Dapi (blue). Graph: Percent of cells with co-clustered integrin α4 and MLCK210 (n = 8 WT BSA; n = 13 WT SDF; n = 8 *Mlck210*[−/−] BSA; n = 4 *Mlck210*[−/−] SDF where n = experiments). p = 0.6999 WT BSA vs *Mlck210*[−/−] BSA; p < 0.0001 WT SDF vs *Mlck210*[−/−] SDF. Statistical significance determined by one-sided Anova with Tukey's multiple comparisons. Scale bars = 5 μm. Data are presented as mean values ± SEM. Experiments were performed twice except for mass spectrometry, which was performed once. Source data are provided with this article as a Source Data file.

determine whether MLCK210 is required for integrin α4-mediated adhesion, *Mlck210*[−/−] and WT murine peripheral blood CD11b+ myeloid cells were incubated with basal medium or medium containing IL-1β, IL-6, SDF-1α, TNFα, VEGF-A, or Lewis lung carcinoma cell conditioned medium (TCM). Cells were then allowed to adhere to endothelial cell monolayers (HUVEC) or to recombinant soluble VCAM-1 coated on plastic plates. While cytokine/chemokine-stimulated WT myeloid cells adhered to HUVEC and VCAM-1, *Mlck210*[−/−] myeloid cells did not adhere (Fig. 2b). Previously, we showed that integrin α4-mediated adhesion of Gr1[lo] monocytes and Gr1[hi] granulocytes to endothelium required the activity of both PI3Kγ and Rap1[7,9]. As both CD11b+ Gr1[lo] monocytes and CD11b+ Gr1[hi] granulocytes extravasate into tumors from peripheral blood, we asked whether the adhesion of each of these two cell types was similarly regulated by PI3Kγ, Rap1, and MLCK210. We found that the adhesion (MFI) of both CD11bGr1[lo] monocytes and CD11bGr1[hi] granulocytes from *Pik3cg*[−/−] (*p110γ*[−/−]), *Mlck210*[−/−], or *Rap1a*[−/−] mice to VCAM-1-coated plates was strongly inhibited compared to the adhesion of cytokine-stimulated WT cells (Fig. 2c). In support of these findings, *Mlck* siRNA-treated myeloid cells also exhibited poor adhesion to HUVEC and VCAM, in contrast to control All Stars siRNA-transfected and untransfected cells (Fig. 2d). These results indicate that MLCK210 plays a key role in the PI3Kγ-Rap1 pathway that promotes integrin α4-mediated adhesion.

To determine whether pharmacological inhibitors of MLCK could suppress cytokine/chemokine-mediated integrin activation and adhesion, WT murine myeloid cells were incubated with medium, SDF-1α, or IL-1β in the presence or absence of either **MW01-022AZ**[23] or **ML-7**, two MLCK inhibitors. We found that MLCK inhibitors suppressed SDF-1α and IL-1β induced adhesion of murine myeloid cells to VCAM-1 in a dose-dependent manner (Fig. 2e, Supplementary Fig. 3a). In contrast, inhibition of MLCK210 did not affect adhesion of myeloid cells to other substrates (Supplementary Fig. 3b). **MW01-022AZ** also inhibited cytokine/chemokine-stimulated human myeloid cell adhesion to HUVEC or VCAM (Fig. 2f). As cell adhesion is a pre-requisite for cell migration and invasion, we evaluated the effect of MLCK inhibition on myeloid cell migration. MLCK knockdown by siRNA transfection as well as the MLCK inhibitor **ML-7** suppressed myeloid cell migration in transwell assays, while control transfection or vehicle had no effect on cell migration on VCAM-1 coated surfaces (Fig. 2g; Supplementary Fig. 3c, d).

In contrast to integrins on cells within tissues, integrins on the surface of circulating immune cells reside in stable, inactive conformations that are unable to bind ligand or facilitate adhesion[5,12]. Upon exposure to cytokines or chemokines, intracellular signal transduction events promote integrin unfolding and acquisition of ligand-binding and adhesive capacity within 0.1 s, an event that is termed integrin activation[12,24]. We previously showed that recombinant VCAM-1 conjugated to the Fc portion of the immunoglobulin heavy chain could bind to integrin α4β1 on cytokine-stimulated myeloid cells but not on unstimulated myeloid cells or on myeloid cells with inactivating mutations in integrin α4, PI3Kγ, Ras, or Rap1a[7,9]. To determine if MLCK210 also regulates integrin α4β1 ligand binding, we incubated human peripheral blood myeloid cells with the MLCK inhibitor **MW01-022AZ** or vehicle and then with recombinant VCAM-1-Fc and anti-Fc fluorochrome-conjugated antibody and quantified the mean fluorescence intensity of bound VCAM. We found that cytokine/chemokine stimulation induced VCAM-1 binding, while the MLCK inhibitor **MW01-022AZ**, as well as the inhibitor **ML-7**, suppressed VCAM-binding (Fig. 3a, b; Supplementary Fig. 3e). Mn2+, which activates integrins independently of cytokine/chemokine receptor signaling by establishing salt bridges between integrins and their ligands[25,26], stimulated VCAM-Fc binding independently of cytokine stimulation (Fig. 3a, b; Supplementary Fig. 3e). **MW01-022AZ** also inhibited binding of HUTS21, a monoclonal antibody that selectively binds to an epitope that is only exposed on the β1 subunit of heterodimeric integrin complexes of cytokine/chemokine-stimulated human immune cells (Fig. 3c, d). In contrast, **MW01-022AZ** had no effect on the binding of P4C10, an anti-β1 integrin antibody that binds independently of activation (Fig. 3c, d). MLCK inhibition also suppressed VCAM-Fc binding to murine myeloid cells, as *Mlck210*[−/−], *Mylk* siRNA-transfected and **MW01-022AZ**-treated myeloid cells failed to bind VCAM-Fc upon cytokine/chemokine stimulation (Fig. 3e–h). Importantly, MLCK210 depletion had no impact on integrin α4 cell surface expression levels. However, we observed a 5–10% decrease in total β1 expression levels in *Mlck210*[−/−] cells (Supplementary Fig. 4a, b). The integrin β1 subunit can partner with over 10 distinct α subunits and is expressed well in excess of any α4 subunit; therefore, these results indicate that *Mlck210* deletion does not affect integrin α4 cell surface expression and does regulate α4 integrin activation. As we previously showed that VCAM-Fc binding to myeloid cells strictly depends on integrin α4β1 expression and function[6,7],

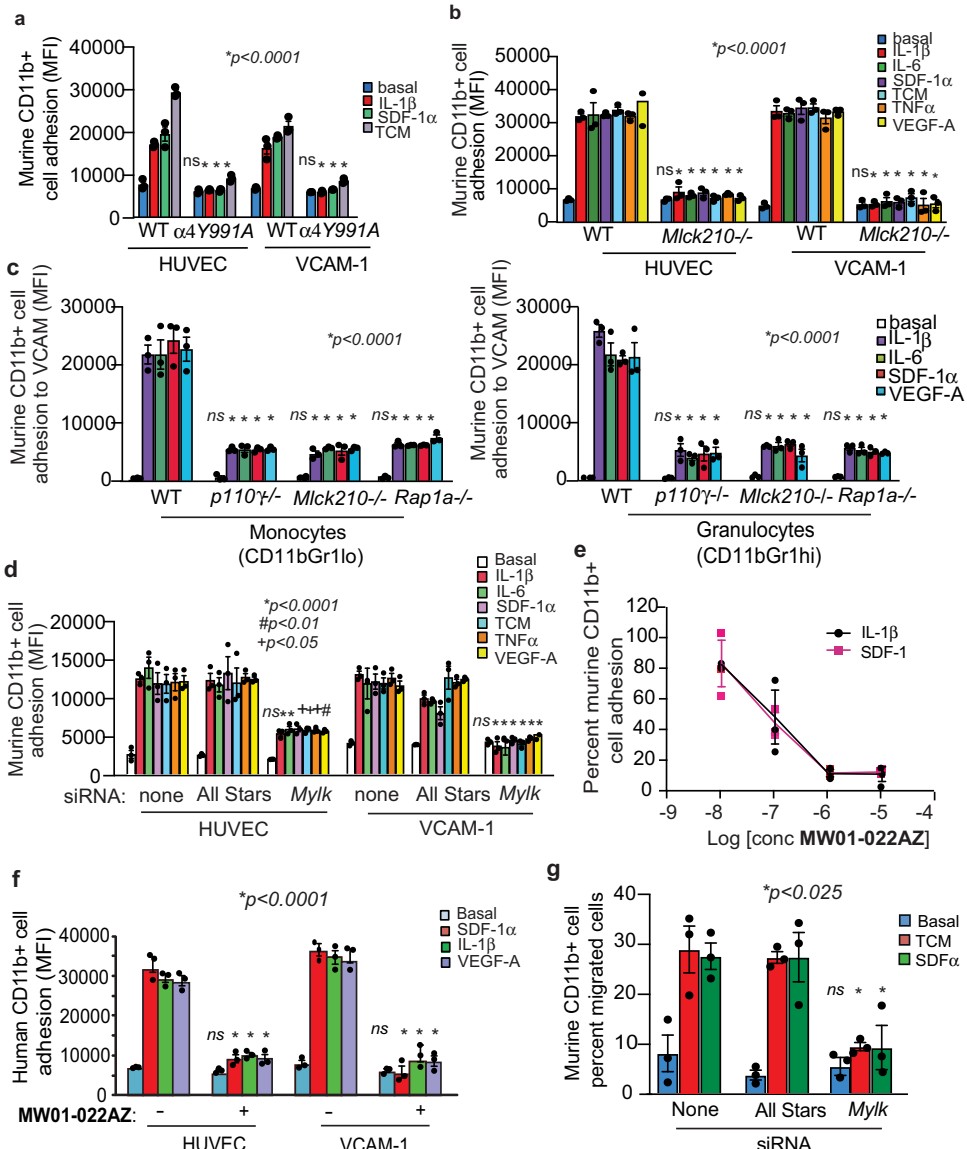

**Fig. 2 MLCK210 promotes integrin α4-mediated myeloid cell adhesion. a** Adhesion to HUVEC or VCAM-1, expressed as mean fluorescence intensity (MFI), of cytokine- and tumor conditioned medium (TCM)-stimulated murine WT or integrin α4Y991A myeloid cells ($n = 3$). *$p < 0.0001$ for cytokine-stimulated WT vs. α4Y991A cells. No significant difference (ns) for adhesion to HUVEC ($p > 0.9999$) or VCAM-1 ($p = 0.9465$) of basal WT vs. α4Y991A cells. **b** Adhesion to HUVEC or VCAM of chemokine, cytokine- and tumor conditioned medium (TCM)-stimulated murine WT or $Mlck210^{-/-}$ myeloid cells ($n = 3$). *$p < 0.0001$ for cytokine-stimulated WT vs $Mlck210^{-/-}$ myeloid cells. No significant difference (ns, $p > 0.9999$) for basal WT vs. $Mlck210^{-/-}$ cells. **c** Adhesion to purified VCAM-1 coated plates of murine CD11bGr1$^{lo}$ monocytes and CD11bGr1$^{hi}$ granulocytes from WT, $p110γ^{-/-}$, $Mlck210^{-/-}$, or $Rap1a^{-/-}$ mice ($n = 3$). *$p < 0.0001$ for cytokine-stimulated WT vs $p110γ^{-/-}$, $Rap1a^{-/-}$, and $Mlck210^{-/-}$ cells. No significant difference (ns, $p = 0.9999$) for basal WT vs $p110γ^{-/-}$, $Rap1a^{-/-}$, and $Mlck210^{-/-}$. **d** Adhesion to HUVEC or VCAM of basal or cytokine-stimulated All Stars or Mlck (Mylk) siRNA-transfected murine myeloid cells ($n = 3$). *$p < 0.0001$ for cytokine induced adhesion of Mlck vs All Stars siRNA-transfected cells to VCAM. *$p < 0.0001$ for adhesion of IL1β and IL-6 stimulated, *$p = 0.0234$ for SDF-1α, *$p = 0.0203$ for TCM, *$p = 0.0133$ for TNFα and *$p = 0.0091$ for VEGF-A stimulated Mlck vs All Stars siRNA-transfected cell adhesion to HUVEC. No significant difference (ns, $p > 0.9999$) for basal Mlck vs All Stars siRNA-transfected cells. **e** Dose response analysis of the effect of **MW01-022AZ** on murine myeloid cell adhesion to VCAM-1 stimulated by SDF-1α or IL1β ($n = 3$). **f** Effect of **MW01-022AZ** on cytokine-stimulated human myeloid cell adhesion to HUVEC or VCAM-1 ($n = 3$). *$p < 0.0001$ for vehicle vs **MW01-022AZ** treated, SDF-1α IL1β and VEGF-A stimulated cells. No significant difference (ns) for basal vehicle vs **MW01-022AZ** on HUVEC ($p = 0.8530$) or VCAM-1 ($p = 0.9666$). **g** Chemotaxis of Mlck or All Stars siRNA-transfected murine myeloid cells toward basal, SDF-1α or tumor cell conditioned medium (TCM) stimulated cells ($n = 3$). *$p = 0.0212$ for TCM and *$p = 0.0189$ for SDF-1α-stimulated All Stars vs Mlck siRNA. No significant difference (ns, $p = 0.9999$) for basal cells. **a**, **d**, **f**, **g** Significance determined by one-way Anova with Tukey's multiple comparisons. All replicates are biologically independent samples. Data are presented as mean values ± SEM. Experiments were performed twice. Source data are provided with this article as a Source Data file.

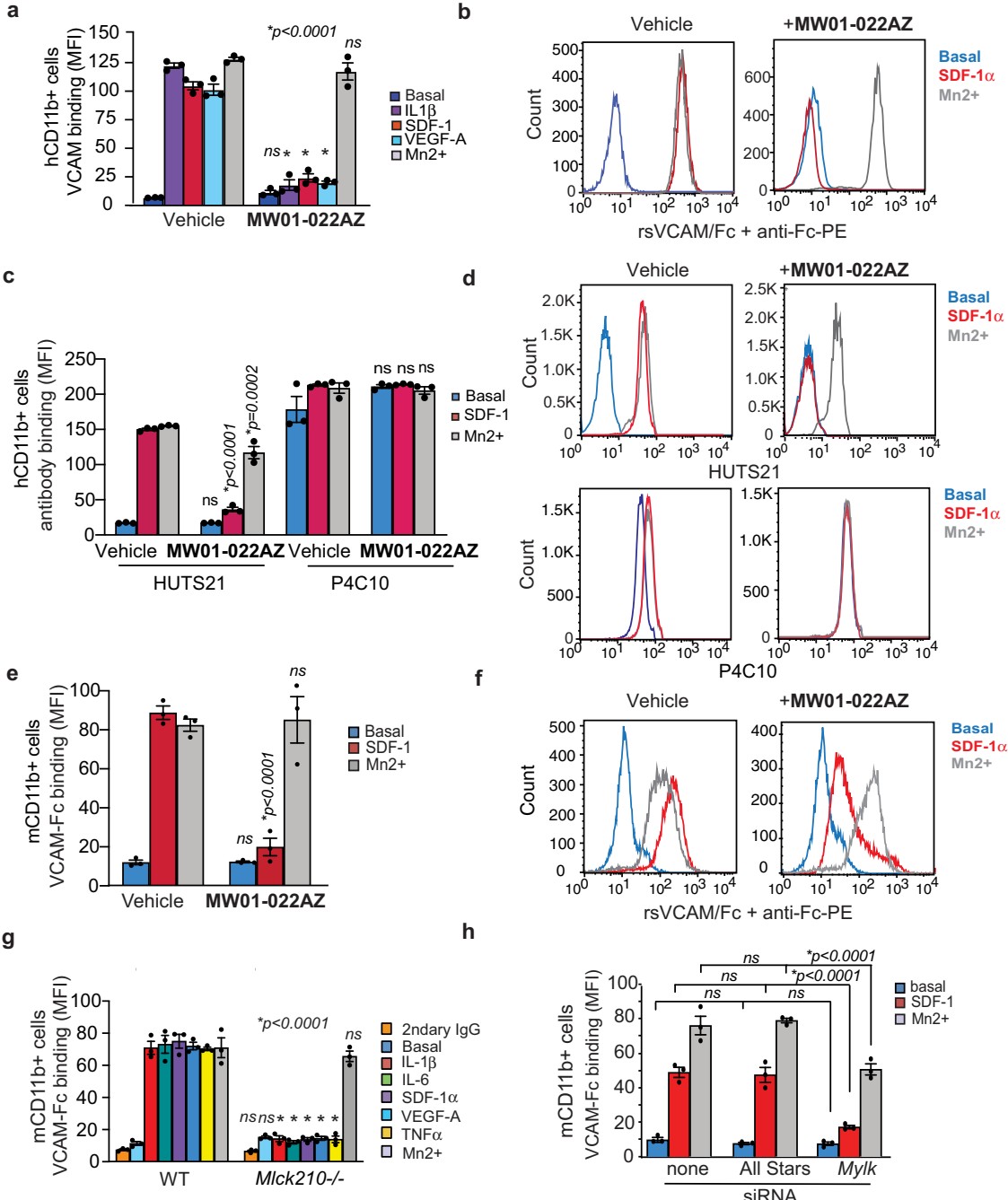

these results show that MLCK210 promotes integrin α4β1 ligand binding (activation) in murine and human myeloid cells.

**MLCK210 is required for Rap1 activation**. Our previous studies showed that PI3Kγ promotes integrin activation and adhesion by activating Rap1[7]. To determine if MLCK210 plays a role in integrin activation and adhesion upstream of PI3Kγ, we analyzed Akt phosphorylation in response to cytokine/chemokine stimulation in WT and $Mlck210^{-/-}$ myeloid cells. Deletion of MLCK210 had no effect on Akt phosphorylation (Fig. 4a). These results suggest that MLCK210 acts downstream of PI3Kγ. We next asked whether MLCK210 could play a role in Rap1 activation, as Rap1 activation is a late step in the signaling cascade leading to integrin activation[27]. Indeed, we found that Rap1 activation, as measured by GTP loading, was impaired in SDF-1α-stimulated $Mlck210^{-/-}$ but not WT myeloid cells (Fig. 4b). In

previous studies, we observed that expression of activated Ras, PI3Kγ, and Rap1 promoted integrin activation and adhesion in the absence of cytokine stimulation[7]. Importantly, we found that expression of activated Ras (RasG12V) or PI3Kγ (p110γCAAX) constitutively activated adhesion of WT but not $Mlck210^{-/-}$ myeloid cells, suggesting that MLCK210 is required for myeloid cell adhesion downstream of Ras and PI3Kγ (Fig. 4c; Supplementary Fig. 5a). In contrast, expression of activated RapG12V stimulated α4β1-mediated adhesion in WT and in $Mlck210^{-/-}$ myeloid cells (Fig. 4c; Supplementary Fig. 5a), indicating that MLCK210 promotes integrin activation downstream of Ras and PI3Kγ and upstream of Rap.

**Myosin promotes integrin activation**. Since the major known substrate for all MLCK isoforms is myosin light chain (MLC), we investigated a role for myosin in integrin α4β1 activation. First,

**Fig. 3 MLCK210 inhibitor prevents integrin α4 activation. a** Quantification of VCAM-Fc binding to IL-1β, SDF-1α, VEGF-A, or Mn2+ stimulated vehicle or MW01-022AZ treated human CD11b+ myeloid cells ($n = 3$). *$p < 0.0001$ for IL-1β, SDF-1α, and VEGF-A stimulated, **MW01-022AZ** vs vehicle-treated cells. No significant difference (ns) for basal ($p = 0.9965$) or Mn2+ ($p = 0.6015$) stimulated **MW01-022AZ** vs vehicle-treated cells. **b** Flow cytometry profiles of cells from **a**. **c** Quantification of HUTS21 or P4C10 antibody binding to basal medium, SDF-1α or Mn2+ stimulated, vehicle or **MW01-022AZ** treated human CD11b+ cells ($n = 3$).*$p < 0.0001$ for HUTS21 binding to SDF-1α stimulated and *$p = 0.0002$ to Mn2+-stimulated **MW01-022AZ** vs vehicle-treated cells. No significant difference (ns) for basal **MW01-022AZ** vs vehicle cells ($p > 0.9999$). No significant difference (ns) for P4C10 binding to vehicle vs **MW01-022AZ** treated basal ($p = 0.1384$), SDF-1α ($p > 0.9999$) or Mn2+ ($p = 0.9997$) stimulated cells. **d** FACs profiles of cells from **c**. **e** Quantification of VCAM-Fc binding to SDF-1α or Mn2+ stimulated murine CD11b+ myeloid cells in **MW01-022AZ** and Vehicle-treated cells ($n = 3$). *$p < 0.0001$ for SDF-1α stimulated cells. No significant difference (ns) for basal ($p > 0.9999$) or Mn2+ ($p = 0.9991$)-treated cells. **f** FACs profiles of cells from **e**. **g** VCAM-Fc binding (MFI) to basal medium, SDF-1α IL-1β, IL-6, VEGF-A, TNFα, and Mn$^{2+}$ stimulated murine myeloid cells from WT or $Mlck210^{-/-}$ mice ($n = 3$). *$p < 0.0001$ for WT vs $Mlck210^{-/-}$ SDF-1α IL-1β, IL-6, VEGF-A, TNFα-stimulated cells. No significant differences (ns) for secondary antibody alone ($p > 0.9999$), basal ($p = 0.9997$), or Mn2+ ($p = 0.9928$) treated cells. **h** VCAM-Fc binding (MFI) to basal medium, SDF-1α and Mn$^{2+}$ stimulated murine myeloid cells from control *All stars* or *Mlck* siRNA-transfected cells ($n = 3$). *$p < 0.0001$ for SDF1α and Mn2+ stimulated *Mlck* siRNA vs All Stars transfected cells. No significant differences (ns) for basal *Mylk* siRNA vs All Stars transfected ($p > 0.9999$) or untransfected vs All Stars transfected cells ($p = 0.9998$). For panels **a**, **c**, **e**, **g**, **h**, significance determined by one-way Anova with Tukey's multiple comparisons. Analysis of VCAM-Fc, HUTS, or P4C10 binding was by single color flow cytometry. All replicates are biologically independent samples. Experiments were performed twice. Data are presented as mean values ± SEM. Source data are provided with this article as a Source Data file.

we found that MLC was phosphorylated in WT cells but not in $Mlck210^{-/-}$ cells (Fig. 4d). Next, we found that MLCK210 catalytic activity was required for MLC phosphorylation in chemokine-treated myeloid cells. While SDF-1 stimulated strongly MLC phosphorylation in WT but not $Mlck210^{-/-}$ myeloid cells, only expression of Flag-tagged intact MLCK210 (WT) but not Flag-tagged kinase-dead MLCK210 (KD) or control DNA (pcDNA) promoted SDF-1-induced MLC phosphorylation in $Mlck210^{-/-}$ cells (Fig. 4e, f). MLCK210 catalytic activity was also required for integrin α4β1-mediated adhesion to VCAM, since expression of catalytically active MLCK210 (WT), but not kinase-dead MLCK210 (KD), restored cytokine/chemokine-stimulated adhesion to VCAM-1 in $Mlck210^{-/-}$ cells (Fig. 4g). Taken together, these results indicate that MLCK210 acts downstream of PI3Kγ but upstream of Rap1 and plays a key role in promoting integrin α4 activation. MLCK can be activated by the influx of calcium ions, resulting in calmodulin-dependent MLCK activation[20]. To explore how MLCK210 becomes activated by PI3Kγ signaling, we evaluated the effects of the PKC inhibitor **R0-32-0432** and the calcium/calmodulin inhibitor W7 on cytokine-mediated adhesion to VCAM. We found that adhesion was suppressed by W7 but not the PKC inhibitor (Supplementary Fig. 5b), indicating a role for calmodulin in MLCK210-dependent integrin activation.

As MLCK210 expression in myeloid cells is required for myosin light chain phosphorylation and for Rap1 activation, our results suggest myosin could play a critical role in integrin α4β1 activation. In fact, myosin has recently been suggested to serve as a scaffold for anchoring signaling proteins at the membrane[28]. We found that primary murine myeloid cells express non-muscle myosin heavy chain MHCIIA (encoded by Myh9) but not MHCIIB (encoded by Myh10) or MHCIIC (encoded by Myh14) protein and mRNA (Fig. 5a, b). Myh9 siRNA, but not Myh10 siRNA (used here as a negative control siRNA), inhibited Myh9 expression (Fig. 5c), and prevented cytokine-stimulated integrin α4 VCAM-Fc binding in SDF-1α but not Mn2+-stimulated myeloid cells (Fig. 5d–e). Similarly, Myh9 siRNA but not Myh10 siRNA prevented cytokine-stimulated myeloid cell adhesion to VCAM-1-coated substrates (Fig. 5f), which is strictly integrin α4β1 dependent[6]. Myh9 siRNA, but not control Myh10 siRNA, also prevented Rap1 GTP loading in SDF-1α stimulated myeloid cells (Fig. 5g). Importantly, blebbistatin, a myosin-selective ATPase inhibitor[29], suppressed cytokine and activated PI3Kγ (p110γCAAX) mediated stimulation of myeloid cell adhesion to VCAM-1, indicating that myosin catalytic activity promotes α4β1-mediated cell adhesion (Fig. 5h). We also observed that

integrin α4 (green) colocalized with myosin (red) in SDF-1α bead-stimulated myeloid cells in $Mlck210^{-/-}$ myeloid cells expressing WT but not kinase-dead MLCK210, indicating that MLCK210 catalytic activity promotes myosin-integrin interactions (Fig. 5i; Supplementary Fig. 5c). Catalytically active MLCK210 was also required to promote chemokine-induced colocalization of α4β1 and Rap1 (Fig. 5i; Supplementary Fig. 5c). Together, these results indicate that MLCK210 phosphorylates and activates myosin, leading to clustering of adaptor and cytoskeletal proteins that are required to activate integrin.

**MLCK210 mediates tumor inflammation and progression.** Since MLCK210 is required for integrin α4β1 activation in myeloid cells and myeloid cell infiltration of tumors is integrin α4β1 dependent[6–8,10], we speculated that $Mlck210^{-/-}$ myeloid cells would exhibit defects in trafficking into sites of tumor inflammation in vivo. $Mlck210^{-/-}$ myeloid cells, like $p110γ^{-/-}$, $Rap1a^{-/-}$, and integrin $α4Y991A$ myeloid cells[6–8,10], exhibited defective recruitment to Lewis lung carcinoma tumors (LLC) but not control tissues, such as spleens, after adoptive transfer into animals with tumors (Fig. 6a–c). Short-term myeloid cell recruitment to LLC tumors was also suppressed when $Mylk$ was knocked down by $Mylk$ siRNA expression in adoptively transferred myeloid cells (Fig. 6d, e). LLC tumor progression as measured by tumor volume over time and endpoint weights were substantially impaired in $Mlck210^{-/-}$ mice compared to WT mice (Fig. 6f–i), as were tumor volumes and weights in mice bearing pancreatic ductal adenocarcinomas (PDAC) (Fig. 6j–l). To evaluate the effect of $Mlck210$ deletion on the tumor microenvironment, we performed flow cytometry to analyze myeloid cell content in $Mlck210^{-/-}$ and WT tumors (Fig. 6m, n). The accumulation of CD11b+ Ly6C+ monocyte/macrophages as well as CD11b+ Ly6G+ granulocytes in tumors was suppressed in $Mlck210^{-/-}$ mice (Fig. 6m–o; see Supplementary Fig. 6a for FACS gating strategy). As monocyte and granulocyte recruitment to tumors was suppressed in $Mlck210^{-/-}$ bone marrow, these results suggest that loss of myeloid cell MLCK210 leads to improved immune responses in tumors.

In prior studies, we used bone marrow transplantation to demonstrate that immune cell PI3Kγ and integrin α4 are both necessary for tumor inflammation and growth[6,7]. We performed similar bone marrow transplantation studies to determine whether immune cell MLCK210 promotes tumor growth in vivo (Fig. 7a). WT mice were irradiated, and bone marrow was reconstituted with either WT or $Mlck210^{-/-}$ bone marrow;

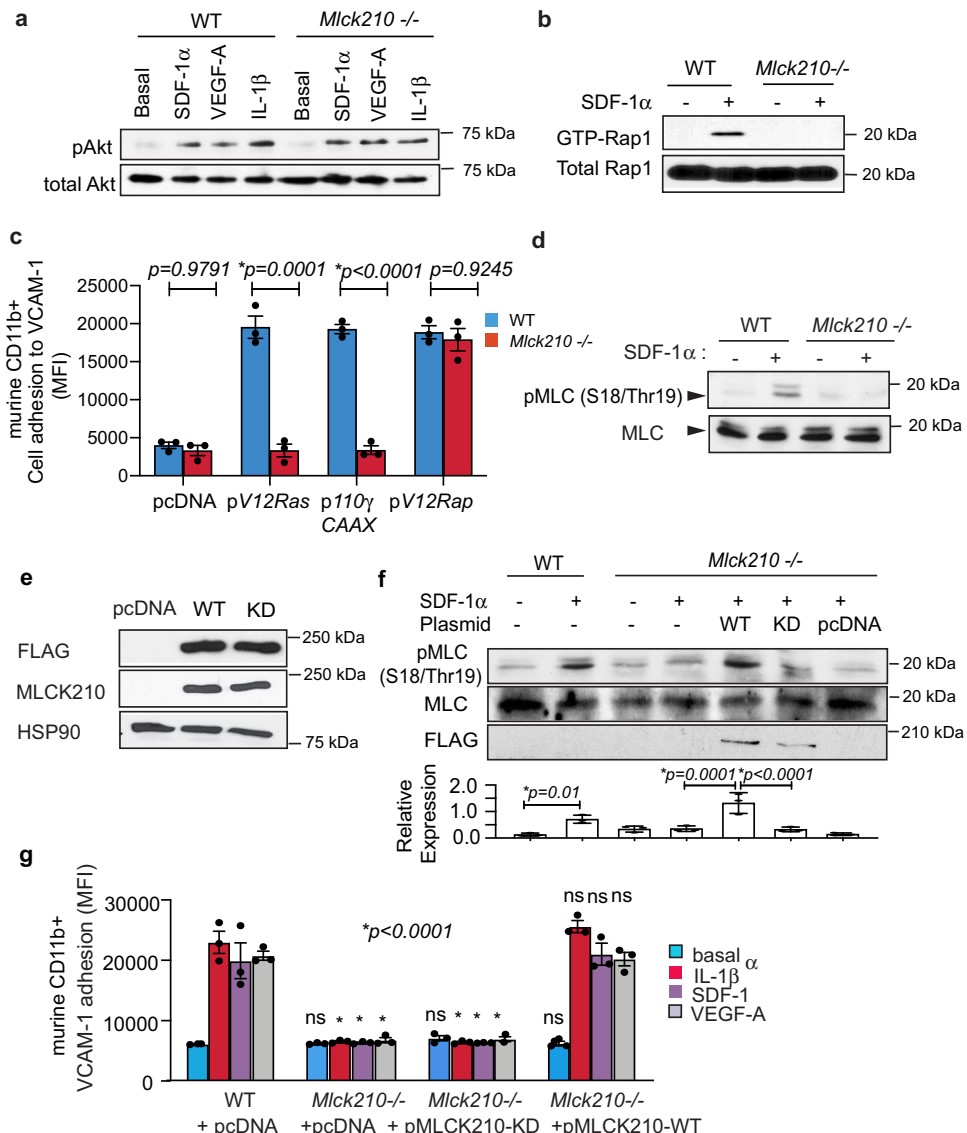

**Fig. 4 PI3Kgamma activates MLCK210 and Rap-mediated adhesion. a** Akt phosphorylation in WT and $Mlck210^{-/-}$ cytokine-stimulated murine myeloid cells. **b** Rap1 GTP loading in cytokine-stimulated WT and $Mlck210^{-/-}$murine myeloid cells. **c** Adhesion to VCAM of WT and $Mlck210^{-/-}$ murine myeloid cells ($n = 3$) after transfection with active Ras ($G12VRas$), active PI3Kγ ($p110\gamma CAAX$), or active Rap ($RapG12V$). $p = 0.9791$ untransfected WT vs $Mlck210^{-/-}$, *$p < 0.0001$ $G12VRas$ and $p110\gamma CAAX$ WT vs $Mlck210^{-/-}$, $p = 0.9245$ $RapV12$ WT vs $Mlck210^{-/-}$. Significance testing by unpaired $t$-test. **d** Myosin Light Chain (MLC) S18/Thr19 phosphorylation in basal or cytokine-stimulated WT and $Mlck210^{-/-}$ murine myeloid cells. **e** Immunoblotting to detect Flag-tagged MLCK210, total MLCK210 and HSP90 in WT and $Mlck210^{-/-}$ murine myeloid cells transfected with control (pcDNA) and MLCK210 expression plasmids. **f** Immunoblotting to detect phosphorylated MLC and total MLC in SDF1a stimulated or unstimulated WT or $Mlck210^{-/-}$ myeloid cells that had been transfected with control (pcDNA) or flag-tagged WT and Kinase-dead (KD) MLCK210 expression plasmids. Graph: Relative pS18/Thr19-MLC expression in 3 independent samples. **g** Cytokine-stimulated VCAM-1 adhesion of WT and $Mlck210^{-/-}$ myeloid cells transfected with control (pcDNA), WT or Kinase Dead (KD) MLCK210 expression constructs ($n = 3$). *$p = 0.0001$ for cytokine-stimulated $Mlck210^{-/-}$ cells and $Mlck210^{-/-}$ cells expressing $Mlck210$KD cDNA vs WT cells. No significant differences (ns) for WT vs $Mlck210^{-/-}$ cells expressing $Mlck210$ WT cDNA, $p = 0.9936$ for IL1β and $p > 0.9999$ for SDF-1a and VEGF-A treated cells. No significant differences (ns) for basal conditions ($p > 0.9999$). Significance determined by unpaired $t$-test for 4c and by one-way Anova with Tukey's multiple comparisons for panels **e**, **g**. All replicates are biologically independent samples. Experiments were performed twice. Data are presented as mean values ± SEM. Source data are provided with this article as a Source Data file.

reconstitution was verified by assessing $Mlck210$ genotypes in DNA isolated from peripheral blood leukocytes (Fig. 7b). Tumor growth, as measured by gross morphology, volume, and weight, was suppressed in mice reconstituted with $Mlck210^{-/-}$ but not WT bone marrow (BM) (Fig. 7c–e). Residual tumors in mice transplanted with $Mlck210^{-/-}$ BM were less than 75 mm³.

To evaluate the effect of $Mlck210$ deletion on the tumor microenvironment, we performed flow cytometry and immuno-fluorescence analysis of myeloid cell, endothelial cell, and T cell

content in tumors from $Mlck210^{-/-}$ and WT BM transplanted tumors. Using flow cytometry, we determined that the major effect of MLCK210 loss was on recruitment of Gr1lo bone-marrow-derived monocyte/macrophages to tumors (Fig. 7f, g; see Supplementary Fig. 6b for FACS gating strategy). Immunofluor-escence analysis of frozen tumors demonstrated that deletion of $Mlck210$ strongly reduced both macrophage and endothelial cell content in tumors, while increasing CD8+ T cell content in tumors (Fig. 7h, i). Tumors in mice bearing $Mlck210^{-/-}$ bone

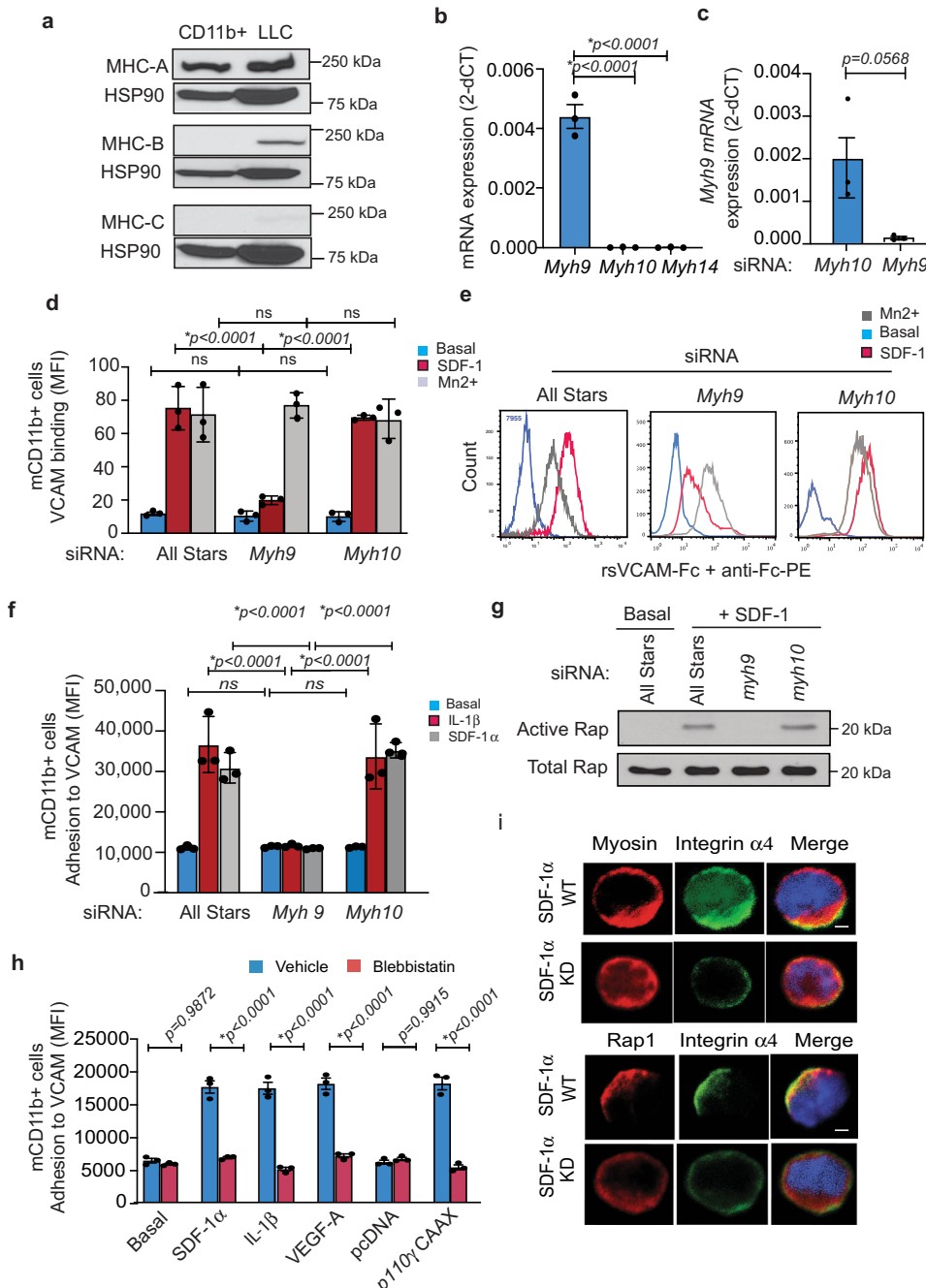

**Fig. 5 MLCK and myosin in myeloid cells. a** Immunoblotting of primary murine myeloid cells (CD11b+) and Lewis lung carcinoma (LLC) cells with anti-myosin heavy chain A, B, and C or HSP90 antibodies. **b** RT-PCR analysis of mRNA expression of *Myh9*, *Myh10*, and *Myh14* in myeloid cells (n = 3, *p < 0.0001 *Myh9* vs *Myh10* or *Myh14*). **c** Effect of *Myh9* and *Myh10* siRNA on *Myh9* expression (n = 3, p = 0.0568). **d**, **e** Quantification (**d**) and FACS profiles (**e**) of VCAM-Fc binding to All Stars, *Myh9* and *Myh10* siRNA-transfected myeloid cells (n = 3). *p < 0.0001 for SDF-1 stimulated, p > 0.9999 for EBM stimulated, and p > 0.9000 for Mn2+ stimulated *Myh9* vs All Stars and vs *Myh10* siRNA-transfected cells. **f** Cell adhesion to VCAM of *Myh9* and *Myh10* siRNA-transfected myeloid cells (n = 3). *p < 0.0001 for SDF-1 and IL-1 stimulated *Myh9* vs All Stars and *Myh10* siRNA. **g** Effect of *Myh9* and *Myh10* siRNA on Rap1 activation by SDF-1α in myeloid cells. **h** Cytokine and p110γCAAX stimulation of myeloid cell adhesion to VCAM-1 in the presence of vehicle or the myosin ATPase inhibitor blebbistatin (n = 3). p = 0.9812 vehicle vs blebbistatin basal medium, *p < 0.0001 vehicle vs blebbistatin for SDF-1α, IL1β, VEGF, p110γCAAX, and p = 0.9915 vehicle vs blebbistatin for pcDNA. **i** Immunofluorescence detection of integrin α4 (green) and myosin or Rap1 (red) in SDF1-bead-stimulated *Mlck210−/−* myeloid cells that were transfected with WT or kinase-dead *Mlck210* cDNA. Nuclei were detected with Dapi (blue). Scale bar = 5 μm. All replicates are biologically independent samples. Data are presented as mean values ± SEM. Significance determined by one-way Anova with Tukey's multiple comparisons for panels **b**, **d**, **f**, and by unpaired *t*-test for panels **c**, **h**. Experiments were performed twice. Source data are provided with this article as a Source Data file.

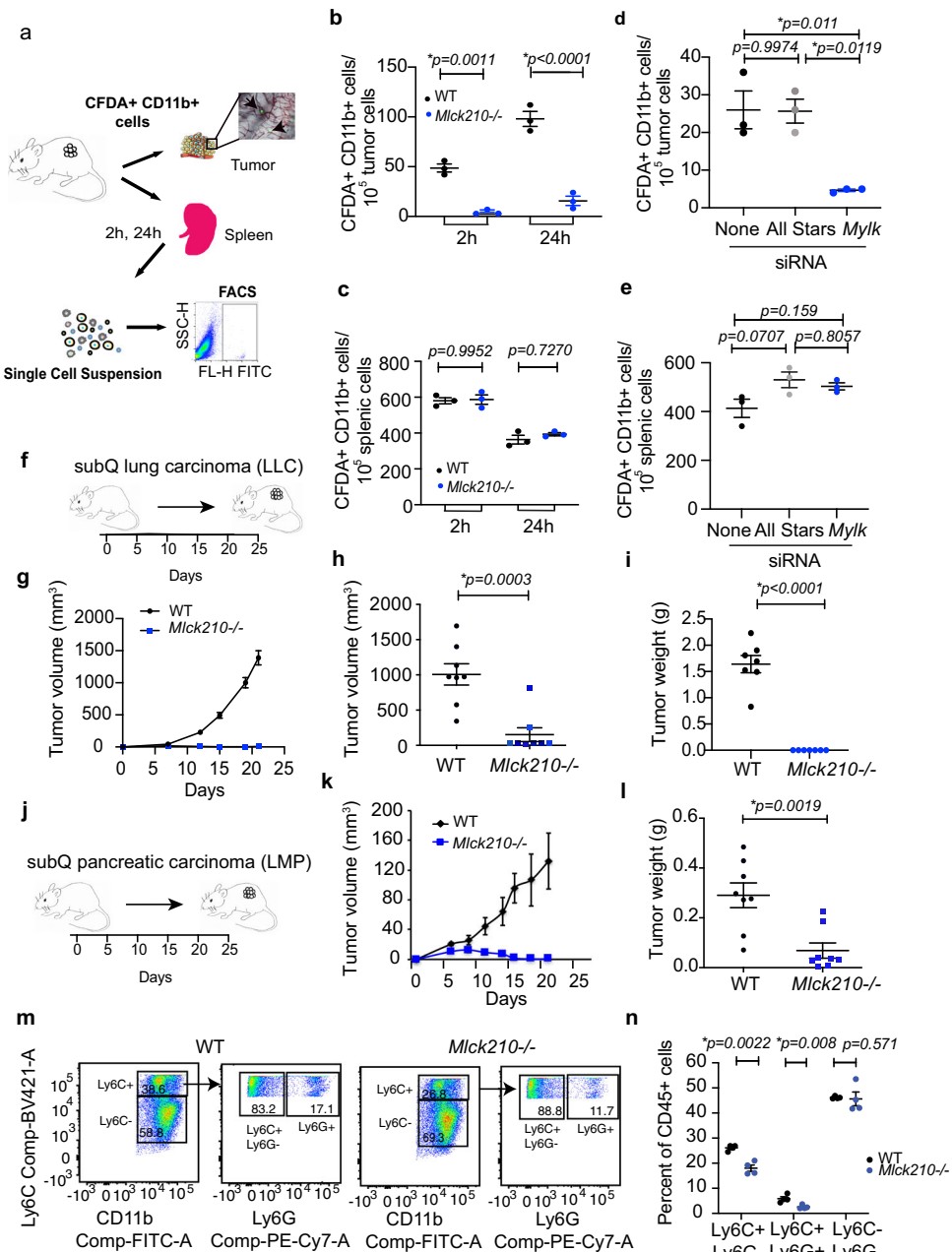

**Fig. 6 MLCK210 deletion inhibits tumor inflammation and tumor growth. a** Schematic of fluorescent myeloid cell adoptive transfer into tumor bearing animals and method of analysis. Quantification of transferred cells was by single color flow cytometry. **b** Number of CFDA-labelled WT and *Mlck210*−/− CD11b+ cells in tumors 2 h or 24 h after adoptive transfer into mice (*n* = 3 mice), *p = 0.0011 2 h and *p < 0.0001 24 h WT vs *Mlck210*−/− by *t*-test. **c** Number of CFDA-labelled WT and *Mlck210*−/− CD11b+ cells in spleens 2 h or 24 h after adoptive transfer into mice (*n* = 3 mice), *p* = 0.9952 2 h and *p* = 0.727 24 h WT vs *Mlck210*−/− by *t*-test. **d** Number of CFDA-labelled control All Stars and *Mylk* siRNA-transfected CD11b+ cells in tumors 2 h after adoptive transfer into mice (*n* = 3 mice), *p* = 0.011 All Stars vs *Mylk* siRNA-transfected by *t*-test. **e** Number of CFDA-labelled control All Stars and *Mylk* siRNA-transfected CD11b+ cells in spleens 2 h after adoptive transfer into mice (*n* = 3 mice). Significance testing by one-way Anova with Tukey's posthoc testing. **f** Schematic of LLC lung tumor implantation and growth. **g–i** Mean ± SEM tumor volume over time (**g**), endpoint volume *p = 0.0003 (**h**), and weight *p < 0.0001 (**i**) of LLC tumors implanted in WT (*n* = 8 mice) and *Mlck210*−/− mice (*n* = 8 mice). **j** Schematic of LMP pancreatic carcinoma lung tumor implantation and growth. **k** Time course of pancreatic ductal adenocarcinoma (LMP) tumor growth in WT and *Mlck210*−/− tumors (*n* = 4 mice). **l** Weight of pancreatic ductal adenocarcinoma tumors from in WT and *Mlck210*−/− tumors (*n* = 4 mice) *p = 0.0019 for WT vs *Mlck210*−/− BM. **m** FACs profiles of myeloid cells in WT and *Mlck210*−/− LLC tumors. For Facs gating scheme see Supplementary Fig. 7a. **n** Quantification of myeloid cells in WT and *Mlck210*−/− LLC tumors (*n* = 4 mice). Significance testing was performed by unpaired two-sample Student's *t*-test. Data are presented as mean values ± SEM. Experiments were performed twice. Source data are provided with this article as a Source Data file.

marrow were strikingly more necrotic than WT tumors (Fig. 7i). As monocyte and macrophage recruitment to tumors was suppressed, while T cell recruitment and necrosis were increased in mice with *Mlck210*−/− bone marrow, these results suggest that

loss of myeloid cell MLCK210 leads to improved immune responses in tumors. To determine whether MLCK210 loss might affect T cells directly, we examine the expression of MLCK210 in T cells isolated from WT and *Mlck210*−/− animals and performed

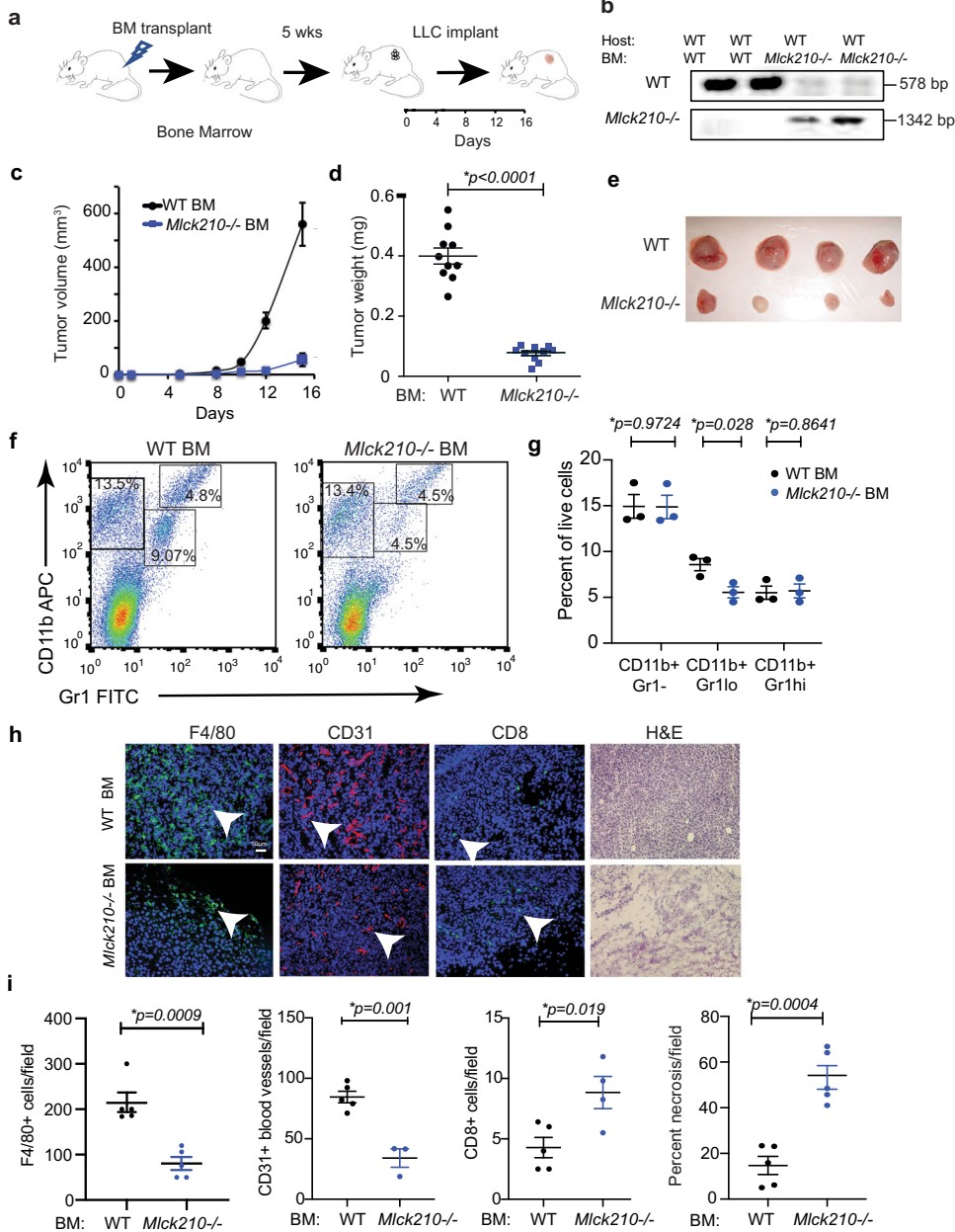

**Fig. 7 Deletion of bone marrow cell MLCK210 inhibits tumor growth. a** Schematic of bone marrow transplantation and tumor implantation. **b** Genomic PCR to detect *Mlck*210 gene in peripheral blood cells from BMT transplanted mice (*n* = 2 mice). **c–e** Tumor volumes (**c**), weights at endpoint (15 days) (*\*p* < 0.0001 WT vs *Mlck210⁻/⁻* BM) (**d**) and images of gross tumor morphology (**e**) from LLC tumors in WT mice transplanted with bone marrow from WT (*n* = 10 mice) or *Mlck210⁻/⁻* mice (*n* = 10 mice). **f** Representative FACS profiles and **g** quantification of Gr1-CD11b+ (*p* = 0.9724) Gr1loCD11b+ (*\*p* = 0.0280), Gr1hiCD11b+ (*p* = 0.8641) myeloid cells in representative WT vs *Mlck210⁻/⁻* BM transplanted tumors harvested at endpoint (15 days) (*n* = 3 mice). For Facs gating scheme, see Supplementary Fig. 7b. **h** Representative histographs and **i** quantification of F4/80+ macrophages/field (*n* = sections from 5 WT mice and 5 *Mlck210⁻/⁻*) *\*p* = 0.0009, CD31+ blood vessels/field (*n* = sections from 5 WT and 3 *Mlck210⁻/⁻* mice) *\*p* = 0.001, CD8+ T cells/field (*n* = sections from 5 WT and 4 *Mlck210⁻/⁻* mice) *\*p* = 0.019 in tumor sections and 100X H&E-stained tumor sections to detect percent necrosis/field (*n* = sections from 5 mice) *\*p* = 0.0004 from mice transplanted with WT vs *Mlck210⁻/⁻* bone marrow. Data are presented as mean values ± SEM. Significance testing was performed by Student's *t*-test. Experiments were performed twice. Source data are provided with this article as a Source Data file.

western blotting to detect MLCK isoforms. We found that T cells only expressed the 108 kDa isoform of MLCK while myeloid cells expressed the 210kD isoform (Supplementary Fig. 7). These results indicate that depletion of MLCK210 in bone-marrow-derived myeloid cells promotes the recruitment of CD8+ T cells into tumors.

To examine further whether MLCK210 in myeloid cells promotes tumor growth, we evaluated the effect of clodronate

and control liposomes on tumor growth in the WT and MLCK210⁻/⁻ background (Fig. 8a). Clodronate is a bisphosphonate that is ingested by macrophages, leading to apoptotic cell death. Clodronate liposomes but not control liposomes reduced tumor growth in the WT background to the same degree as did *Mlck210* deletion (Fig. 8b, c); however, clodronate had no additive inhibitory effect on tumor growth in the *Mlck210⁻/⁻* background (Fig. 8b, c). Macrophages were efficiently depleted from WT and

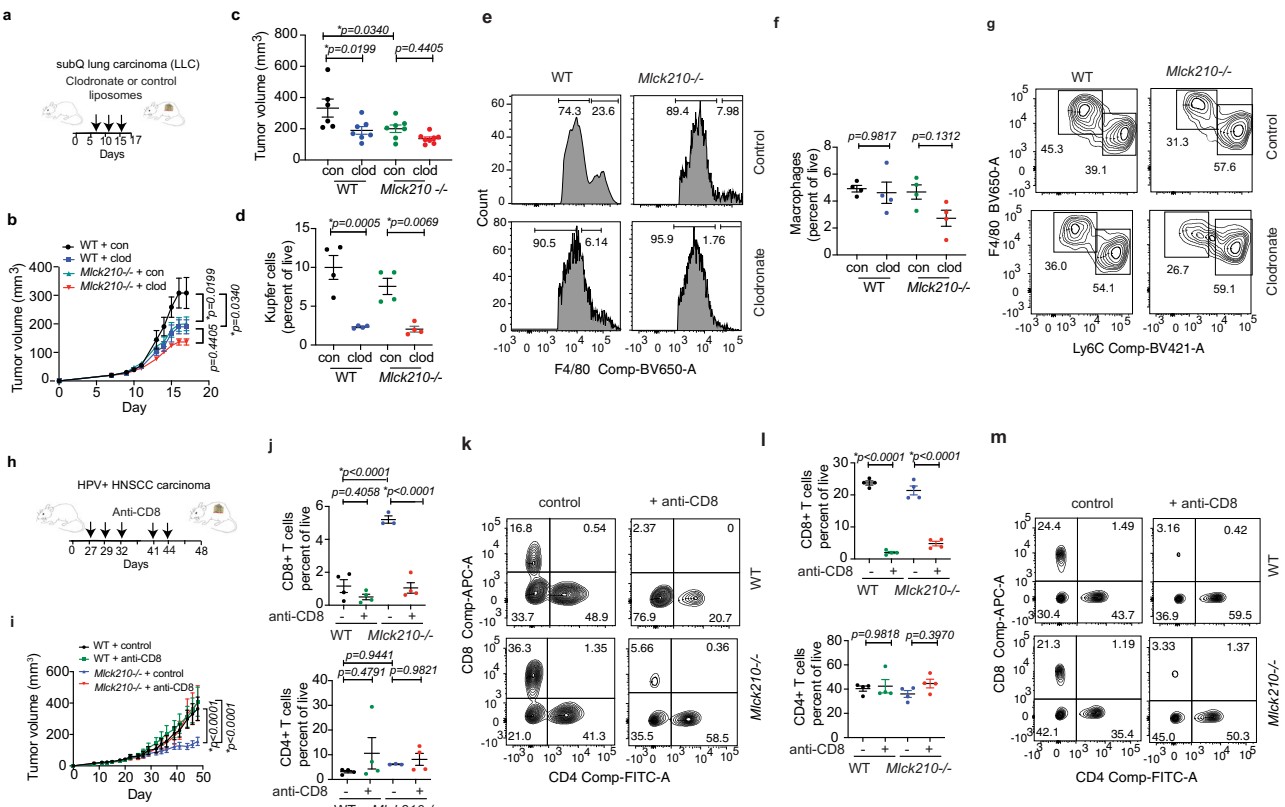

**Fig. 8 CD8+ T cells facilitate *Mlck210*−/−-mediated tumor suppression. a** Schematic of clodronate and control liposome treatment of LLC tumor bearing mice. **b, c** LLC tumor volumes over time (**b**) and endpoint tumor volumes (**c**) for WT (*n* = 6 mice treated with control liposomes (con) and 8 mice treated with clodronate liposomes (clod) and for *Mlck210*−/− [*n* = 7 mice treated with control liposomes (con) and clodronate liposomes (clod)]. One WT animal treated with clodronate liposomes died on day 14 post-tumor inoculation and was not included in the final tumor weight graph. **d–g** Quantification (**d, f**) and FACs plots of Ly6C- (**e**) liver and (**g**) tumor macrophages from WT and *Mlck210*−/− mice after treatment with clodronate and control liposomes (*n* = 4 mice). For Facs gating scheme, see Supplementary Fig. 7a. **h** Schematic depicting anti-CD8 and control antibody treatment of HPV+ Meer WT and *Mlck210*−/− tumor bearing mice. **i** Time course of tumor growth for control and anti-CD8 antibody-treated WT and *Mlck210*−/− mice (*n* = 12 WT and *n* = 8 *Mlck210*−/− control mice, and *n* = 11 WT and *n* = 8 *Mlck210*−/− anti-CD8-treated mice). Statistical significance determined by linear regression analysis. **j, k** Quantification (**j**) and FACs plots (**k**) of tumor associated CD8+ and CD4+ T cells in mice from **i** (*n* = 4 mice). For Facs gating schemes, see Supplementary Fig. 7c. **l, m** Quantification (*n* = 4 mice) and FACs plots (**m**) of splenic CD8+ and CD4+ T cells in mice from **i**. For Facs gating schemes, see Supplementary Fig. 7c. Statistical significance determined by one-way Anova with Tukey's posthoc testing. Data are presented as mean values ± SEM. Experiments were performed twice. Source data are provided with this article as a Source Data file.

*Mlck210* knockout mouse livers (Fig. 8d, e) and variably reduced in tumors (Fig. 8f, g). The results support the conclusion that *Mlck210* deletion, like clodronate liposomes, interferes with the accumulation of tumor-promoting macrophages.

To determine whether recruitment of CD8+ T cells is required for the inhibition of tumor growth by *Mlck210* deletion, we evaluated the effect of CD8+ T cell depletion on tumor growth in WT and *Mlck210*−/− animals (Fig. 8h). Antibody-mediated depletion of CD8+ T cells prevented inhibition of tumor growth in *Mlck210*−/− mice but had little added effect on tumor growth in WT mice (Fig. 8i). We observed that tumors implanted in the *Mlck210*−/− background recruited substantially more (approximately five times more) CD8+ T cells to tumors, while CD4+ T cells content was not significantly different in tumors in *Mlck210*−/− and WT mice (Fig. 8j, k; see Supplementary Fig. 6a for Facs gating strategy). CD8+ T cell antibody treatment effectively removed CD8+ T cells from both spleen and tumors (Fig. 8j–m). Based upon these studies, we concluded that *Mlck210* deletion in myeloid cells promotes to CD8+ T cell recruitment to tumors and subsequent tumor suppression.

Importantly, tumor growth and inflammation were also strongly suppressed in mice treated with the MLCK inhibitor **MW01-022AZ** (Fig. 9a–d). Gr1hi granulocyte, as well as Gr1lo

monocyte/macrophage infiltration, were suppressed in drug-treated tumors (Fig. 9e, f). Importantly, macrophage and endothelial cell content was suppressed, while T cell content was increased in drug-treated tumors (Fig. 9g, h). As we previously documented that tumor associated myeloid cells promote angiogenesis and inhibit T cell recruitment to tumors, these studies support the conclusion that myeloid cell MLCK210 plays a significant role in regulating tumor inflammation and growth in vivo by promoting myosin-mediated integrin activation and myeloid cell recruitment.

## Discussion
Our prior studies showed that pharmacological or genetic blockade of PI3Kγ suppressed integrin α4β1-mediated myeloid cell adhesion and recruitment of monocytes and granulocytes into tumors and inhibited disease progression[6,7,30]. Here we identified high molecular weight myosin light chain kinase, MLCK210, as a key regulator of integrin activation during tumor inflammation and progression. We found that MLCK210 promotes myosin-dependent, Rap1-mediated, integrin α4β1 activation, leading to myeloid cell adhesion to endothelium, tumor inflammation, and tumor progression.

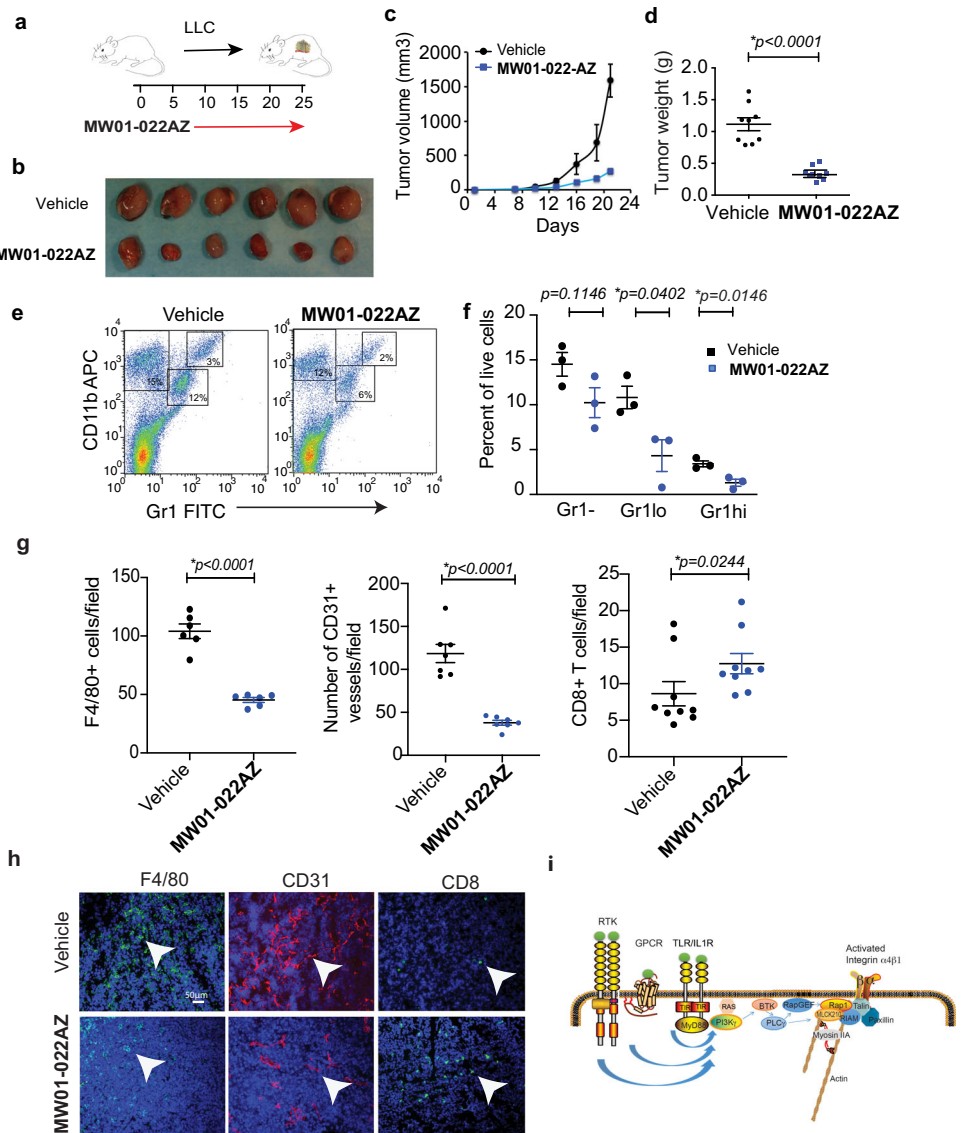

**Fig. 9 Pharmacological inhibitor of MLCK MW01-022AZ suppressed tumor inflammation and growth. a** Schematic of LLC lung tumor implantation in C57Bl6 mice and **MW01-022AZ** treatment. **b–d** Effect of **MW01-022AZ** on LLC **b** gross tumor morphology ($n$ = tumors from 6 mice) **c** tumor volume ($n$ = 9 mice and **d** tumor weight at endpoint (21 days) ($n$ = 9 mice). *$p$ < 0.0001 for vehicle vs **MW01-022AZ** treatment, determined by $t$-test. **e** FACS profiles and **f** quantification of Gr1-CD11b+ ($p$ = 0.1146) Gr1loCD11b+ (*$p$ = 0.0402), Gr1hiCD11b+ (*$p$ = 0.0146) myeloid cells in representative **MW01-022AZ** vs vehicle-treated tumors at endpoint (21 days) ($n$ = 3 mice), determined by $t$-test. For Facs gating scheme, see Supplementary Fig. 7b. **g** Quantification and **h** representative histographs of F4/80+ macrophages ($n$ = sections from 6 mice) *$p$ < 0.0001 for vehicle or **MW01-022AZ** by $t$-test, CD31+ blood vessels ($n$ = sections from 7 mice) *$p$ < 0.0001 for vehicle or **MW01-022AZ** by $t$-test, CD8+ T cells ($n$ = sections from 9 mice) *$p$ = 0.0244 for vehicle vs **MW01-022AZ** by Mann–Whitney test in tissues from mice treated with vehicle or **MW01-022AZ**. **i** Model of integrin activation demonstrating that cytokine stimulation activates a pathway whereby PI3Kγ induces MLCK210 and myosin activation and colocalization with Rap1, resulting in Rap1-mediated integrin α4 conformation changes and interaction of the integrin with the actin cytoskeleton. Data are presented as mean values ± SEM. Statistical significance determined by Student's $t$-test. Experiments were performed twice. Source data are provided with this article as a Source Data file.

Our studies showed MLCK210 colocalizes with integrin α4β1 in the plasma membrane of cytokine and chemokine-stimulated myeloid cells and that integrin α4β1 ligand binding and adhesion in response to cytokine/chemokine stimulation requires MLCK210. We found that Rap1 activation, which is required for integrin α4 activation, is impaired in MLCK210$^{-/-}$ myeloid cells; expression of activated Rap1 restored adhesive functionality to $Mlck210^{-/-}$ cells. We also showed that MLCK210 controls integrin activation downstream of PI3Kγ, a key regulator of tumor inflammation. These results identify a key PI3Kγ-MLCK210-myosin-Rap-integrin α4 molecular pathway that

regulates tumor inflammation (Fig. 9i). As myeloid cells lacking MLCK210 exhibit defects in integrin α4β1 activity despite high expression levels of the non-muscle MLCK108 isoform, these results suggest that the unique N-terminal domain of MLCK210 plays a role in promoting integrin activation. Importantly, inflammation and tumor growth were substantially impaired in $Mlck210^{-/-}$ mice and in mice transplanted with $Mlck210^{-/-}$ bone marrow. These results are similar to those observed in PI3Kγ$^{-/-}$ and integrin α4Y991A mutant mice[6,7,30], indicating that MLCK210 plays a significant role in integrin activation in vivo.

Our studies show that MLCK210 expression in myeloid cells is required for MLC phosphorylation and for Rap1 activation, suggesting that the MCLK substrate myosin could play a critical role in integrin activation. Indeed, knockdown of *Myh9*, the major myosin isoform expressed in myeloid cells, as well as the myosin ATPase inhibitor blebbistatin inhibited integrin activation and adhesion. Support for a role of myosin in integrin activation is also provided by studies showing that integrin α4-myosin interactions promote α4-mediated activation in transfected Chinese Hamster Ovary cells[30] and that myosin IIA-Rap1 interactions can promote Rap1-dependent integrin activation in fibroblasts[31,32].

Activation of integrin α4β1 by inside-out signaling promotes leukocyte extravasation from the vasculature[9,12]. Integrin activation depends on Rap1[6,7] and its effector protein RIAM, which localizes talin to the membrane[33–35]. Talin binds integrin β chain cytoplasmic tails, resulting in an allosteric increase in ligand binding affinity[33–35]. Paxillin binding to the integrin α4 cytoplasmic tail promotes integrin activation, adhesion, and trafficking of lymphocytes and myeloid cells[9–12]. We showed here that integrin α4β1, MLCK210, talin, and paxillin are all colocalized in cytokine-stimulated myeloid cells and that MLCK210 expression is necessary to promote Rap1 and paxillin colocalization with α4 integrin. Our data show that MLCK210 catalytic activity is required to promote integrin clustering and adhesion, as catalytically active MLCK210 but not inactive MLCK210 colocalizes with α4β1 and restores integrin clustering and adhesion to *Mlck210*⁻/⁻ cells. Furthermore, MLCK210 phosphorylates myosin light chain, thereby inducing myosin ATPase activity and integrin activation in myeloid cells. The unique N-terminal head group of MLCK210 has been shown to target this kinase to the membrane and to actin filament[36], suggesting that MLCK210 may promote integrin activation by facilitating colocalization of integrin, signaling proteins, and cytoskeletal components (Fig. 9i).

Integrin cytoplasmic tails play critical roles in integrin activation. Integrin alpha and beta subunit cytoplasmic tails interact with each other through salt bridges between the membrane proximal GFKKR motif of the alpha subunit and the membrane proximal HDRRE motif of the beta chain[37–42]. R→A mutations in the GFFKR sequence in the cytoplasmic tails of α4, αIIb, and αL constitutively activate these integrins[43]. Disruption of this salt bridge forces the cytoplasmic tails apart and alters extracellular domain conformations to increase ligand-binding affinity. The interaction of talin with NPxY sites in integrin β chain cytoplasmic tails also alters the packing of transmembrane domains and activates integrins[43]. Talin binding is facilitated by Rap1 activation, which promotes Rap1 effector protein RIAM to localize talin near the membrane[8]. Paxillin binding to the integrin α4 cytoplasmic tail also enhances integrin activation, as a Y→A mutation of the Y991 paxillin-binding site to A991 reduces talin binding and inhibits adhesion and trafficking of lymphocytes and myeloid cells[6,10–12]. Our studies here show that MLCK210 also play a key role in activating integrin α4β1 by promoting myosin-dependent Rap1 activation and localization and subsequent Talin and Paxillin binding to the integrin cytoplasmic tail, thereby contributing to cancer progression. A previous study described a role for MLCK in β2 integrin activation; those studies described a myosin-independent role for low molecular weight MLCK in beta subunit integrin activation[44]. In contrast, our studies demonstrate a role for a high molecular weight variant of MLCK, MLCK210, in mediating integrin activation, inflammation, and cancer progression.

## Methods

The research reported in this article complies with all relevant ethical regulations. All animal experiments were performed with approval of the Institutional Animal Care and Use Committee of the University of California San Diego. All use of human cells was approved by the Institutional Review Board for human subjects research of the University of California, San Diego.

**Reagents**. **MW01-022AZ** was provided by D.M. Watterson, Northwestern University. Flag-tagged *Mlck210* and Flag-tagged kinase-dead *Mlck210* expression constructs were provided by D.M. Watterson, Northwestern University. *RasV12, RapV12, p110γCAAX* pcDNA plasmids were previously described[43]. All plasmids were verified by DNA sequencing. **ML-7** was from Selleck Chemicals. C57BL/6 LLC cells were obtained from the American Type Culture Collection (ATCC). LMP pancreatic tumor cells were obtained from Dr. Andrew Lowy, UCSD Moores Cancer Center[45]. HPV+ MEER cells were obtained from Dr. John Lee, University of South Dakota. All cells were verified by whole-exome sequencing, RNA sequencing, Western blotting and RT-PCR. All cell lines were negative for mycoplasma and mouse pathogens. All antibodies we used were validated by commercial vendors as determined by representative data shown on product websites and verified by our own analysis based on appropriate patterns of expression in western blotting, protein biochemistry, and knockdowns studies.

**Cell lines**. C57BL/6 Lewis lung carcinoma (LLC) were obtained from ATCC (CRL-1642), LSL-Kras^G12D/+;LSL-p53^R172H/+;Pdx1Cre pancreatic carcinoma (LMP) were obtained from Andrew Lowy, University of California, San Diego, and HPV+ MEER head and neck squamous cell carcinoma cells were obtained from John H. Lee, University of South Dakota. The identity of these cell lines has been extensively validated by RNA sequencing/oncogene profiling, whole-exome sequencing and growth, and protein expression properties.

**Animal studies**. Animal were housed with standard light cycles of 12 h of light and 12 h of darkness, at ambient humidity. Ventilated housing racks provided HEPA-filtered supply and exhausted air to and from the animal cages. *Mlck210*⁻/⁻ male and female animals in the C57BL6 background were from D.M. Watterson, Northwestern University and maintained at the University of California, San Diego. C57BL6 wildtype (WT) male and female 6–8-week-old animals were purchased from Jackson Laboratories. *Pik3cg*⁻/⁻ (p110γ⁻/⁻), *Rap1a*⁻/⁻, and integrin *α4Y991A* mice in the C57BL6 background were maintained at the University of California, San Diego. All animal experiments were performed with approval of the Institutional Animal Care and Use Committee of the University of California San Diego. The maximal tumor burden allowed per mouse at University of California San Diego was 2 cm³. None of our tumor studies were allowed to exceed this tumor burden limitation.

**Human subjects**. Two units of de-identified, pre-existing pooled human peripheral blood cells collected by plasmapheresis by the San Diego Blood Bank were used for human myeloid cell adhesion studies. Blood collection and consenting was performed by the San Diego Blood Bank. This research is not classified as subject to human research regulations but was conducted under approval by the Institutional Review Board for human subjects research of the University of California, San Diego.

**Purification of myeloid cells**. Myeloid cells were purified from murine bone marrow of 6–8-week-old male or female mice. Bone marrow cells harvested from mouse long bones were mixed with RBC Lysis Buffer (ThermoFisher), layered on a 15 ml Histopaque solution and centrifuged at 400 × *g* for 50 min. The mononuclear cell layer was then removed from the gradient, rinsed with PBS and incubated with anti-CD11b affinity magnetic beads, according to the manufacturer's directions (Miltenyi Biotec). To isolate purified monocytes and granulocytes for further study, CD11b+ cells were purified from mouse bone marrow and sorted into CD11b+ Gr1hi and CD11b+ Gr1lo populations by flow cytometry. Human myeloid cells were purified from fresh human peripheral blood cell preparations collected by plasmapheresis by the San Diego Blood Bank by gradient centrifugation and anti-CD11b affinity bead isolation. The purity of the cell populations was assessed by Facs analysis.

**Adhesion assays**. 1 × 10⁵ calcein-AM labelled CD11b+ cells were incubated on 48-well culture plates containing endothelial cell monolayers (HUVEC) or coated with 5 μg/ml recombinant soluble VCAM-1 (R&D Systems), ICAM-1 (R&D Systems), vitronectin, or collagen for 20 min at 37 °C in basal media, culture media from Lewis lung carcinoma cells (TCM) or DMEM containing 200 ng/ml SDF1α, IL-1β, IL-6, TNFα, or VEGF-A (R&D Systems) without or with 1 μM inhibitors directed against MLCK (**ML-7** or **MW01-022AZ**), PKC (Ro-32-0432), calcium/calmodulin (W7), or myosin (Blebbistatin). Additionally, CD11b+ myeloid cells from WT, *Rap1*⁻/⁻, *p110γ*⁻/⁻, and *Mlck210*⁻/⁻ mice as well as *Mlck, Myh9, Myh10*, and non-silencing siRNA-transfected and *RasV12, RapV12*, and *p110γCAAX*-transfected cells were similarly incubated for 20 min on 5 μg/ml rsVCAM-1. After washing three times with warmed medium, adherent cells were quantified using a plate fluorimeter (TECAN). Serum-free Lewis lung carcinoma conditioned medium was prepared by culturing LLC cells in DMEM for 24 h and centrifuging culture medium at 10,000 × *g* to remove debris.

## Migration assays

*Haptotaxis assays.* Eight micrometers of pore size transwell inserts (Costar 3422) were coated on the underside with 5 µg/ml rsVCAM-1 or vitronectin in phosphate buffered saline (PBS) overnight at 4 °C. Inserts were washed and placed into 24-well plates containing 1 ml of DMEM or serum-free tumor conditioned medium (TCM) containing vehicle (0.1% DMSO) or 10 µM **ML-7** in 0.1% DMSO. $5 \times 10^5$ CD11b+ cells from mouse bone marrow in 100 µl of the same medium in the lower chamber were added to the upper chamber, and cells were incubated overnight at 37 °C. Inserts were rinsed in PBS, fixed briefly in 1% paraformaldehyde and then stained for 10 min in 400 µl crystal violet. Inserts were extensively water-washed to remove excess stain. Cells on the upper surface of the insert were removed by wiping with a cotton swab and then migrated cells on the underside of the insert were photographed and counted using an inverted microscope.

*Chemotaxis assay.* Eight micrometers pore size transwell inserts (Costar 3422) were placed into 24-well plates containing basal media, serum-free culture media from Lewis lung carcinoma cells (TCM) or serum-free DMEM containing 200 ng/ml SDF1α. $5 \times 10^5$ CD11b+ cells isolated from mouse bone marrow in 100 µl of serum-free DMEM were added to the upper chamber, and cells were incubated overnight at 37 °C. Inserts were rinsed in PBS, fixed briefly in 1% paraformaldehyde and then stained for 10 min in 400 µl crystal violet. Inserts were extensively water-washed to remove excess stain. Cells on the upper surface of the insert were removed by wiping with a cotton swab and then migrated cells on the underside of the insert were photographed and counted using an inverted microscope.

## Ligand-binding assay

$1 \times 10^6$ WT or $Mlck210^{-/-}$ murine CD11b+ cells or CD11b+ cells transfected with *Mlck, Myh9, Myh10,* or non-silencing siRNAs and myeloid cells treated with **ML-7** or **MW01-022 AZ** were incubated with 200 ng/ml IL-1β1, SDF-1α, IL-6, TNFα, VEGF-A, or medium together with 1 mg/ml mouse VCAM-1/human-Fc fusion protein (R&D Systems #862-VC-100). Cells were washed twice and incubated with donkey anti-human-Fc antibody (Jackson Immunoresearch) for one hour on ice, fixed with 3.7% paraformaldehyde in PBS+ 0.1% BSA and then analyzed by flow cytometry. Mean fluorescence intensity of treated cells was compared to that of unstimulated cells (basal).

## HUT21 integrin activation assay

To quantify integrin activation, $2.5 \times 10^6$ **MW01-022AZ** or control treated human myeloid cells/ml were incubated in culture medium containing 10 µg/ml normal human immunoglobulin (12,000 C, Caltag Laboratories, Burlingame, CA) for 45 min on ice. These cells were then incubated in 200 ng/ml SDF-1α or 1 mM Mn2+ in the presence or absence of 2.5 µg/ml HUTS21 (β1 activation epitope, BD-Bioscience catalog # 556048), P4C10 (total β1, EMD/Millipore # MAB1987), or IgG2 control for 10 min at 37 °C, followed by Alexa 488 goat-anti-mouse antibodies for 20 min on ice. Bound antibody was quantified by flow cytometry.

## siRNA and plasmid transfections

Purified murine CD11b+ cells were transfected with 2 µg/ $1 \times 10^6$ cells pcDNA 3.1 *p110γCAAX* (activated PI3Kγ), pcDNA3.1 R*asV12* (activated Ras), pcDNA3.1 *RapV12* (activated Rap), or *pGFPMax* (GFP) using Lonza Mouse Macrophage Nucleofection Kits and settings. Alternatively, $1 \times 10^6$ CD11b+ cells were transfected with 100 nMoles of validated siRNA targeting *Mylk (Mm_Mylk_1* and Mm_Mylk_2), *Myh9 (Mm_Myh9_1* and *Mm_Myh9_2), Myh10 (Mm_Myh10_1* and *Mm_Myh10_2*), or non-silencing siRNA (Ctrl_AllStars_1) from Qiagen. After transfection, cells were maintained for 24 h in culture media containing 20% serum prior to assessment. Each siRNA was tested individually for efficient knockdown of gene and protein expression and for inhibition of adhesion or ligand binding. To control for off-target effects of siRNAs, only previously validated siRNAs were used.

## Genotyping and gene expression

Genotyping of $Mlck210^{-/-}$ mice was performed using forward primer 5'CTA CTG GTG GAA GCA AGG GAC TG3' and reverse primer 5'CGA GCA CGT ACT CGG ATG GAA G3' on DNA isolated from peripheral blood leukocytes. Total RNA was isolated from primary myeloid cells using RNeasy Kit (Qiagen). cDNA was prepared from 500 ng RNA from each sample and qPCR was performed using primer sets for *Mlck (Mm_Mylk_1_SG), Myh9 (Mm_Myh9_1_SG), Myh10 (Mm_Myh10_1_SG), Myh14 (Mm_Myh14_1_SG)* from Qiagen (QuantiTect Primer Assay). GAPDH (*Gapdh*) sense primers 5'CATGTTCC AGTATGACTCCACTC3' and anti-sense primers 5'GGCCTCACCCCATTTGAT GT3'. Relative expression levels were normalized to *Gapdh* expression according to the formula $[2^{-(Ct\ gene\ of\ interest\ -\ Ct\ gapdh)}]$.

## Rap1 activation assay

Rap activity was measured at 37 °C after stimulation with basal medium or medium containing 200 ng/ml SDF-1α (R&D Systems) for 3 min. GTP-Rap was purified from 1 mg cell lysate by addition of RalGDS-GST fusion proteins and glutathione-conjugated sepharose beads. using a RapGTPase pull-down assay kit (Thermo Scientific #16120). GTP-Rap and total Rap were detected by immunoblotting with anti-Rap1 antibodies including with the kit at 1:1000, according to manufacturer's direction and as previously described[9].

## Immunoprecipitation

Myeloid cells were treated with basal media (DMEM) or serum-free Lewis lung carcinoma conditioned medium for 20 min at 37 °C, rinsed with cold PBS and lysed in Tris-buffered saline containing 1% CHAPS, 20 mM β-glycerophosphate, 1 mM Na3VO4, 5 mM NaF, 100 ng/ml microcystin-LR, and protease inhibitor cocktail. After centrifugation, integrin α4β1 was immunoprecipitated as follows: 1 mg total protein was precleared with 10 µl protein G-conjugated Dynabeads (Thermo Fisher #10007D) for 1 h at 4 °C with rotation. Cleared lysates were incubated with 5 µg of rat anti-mouse integrin α4 (PS2, MilliporeSigma #CBL1304) at 4 °C for 2 h, then with 30 µl of protein G-conjugated Dynabeads for 2 h with rotation. Beads were washed three times with 1 ml cold PBS containing protease inhibitor cocktail. Protein precipitates were electrophoresed on 10% SDS-PAGE gels and immunoblotted. Immunoblot were incubated with anti-integrin α4 (C-20, Santa Cruz Biotechnology #sc-6602 1:1000), anti-paxillin (H-114, Santa Cruz Biotechnology #sc-5574 1:1000), and anti-talin (Clone TD77, EMD Millipore #05-1144 1:1000).

## Immunoprecipitation for peptide sequencing

Bone-marrow-derived monocytic cells were isolated as described and treated with either DMEM or a conditioned medium (TCM) that was freshly collected from the culture of Lewis lung carcinoma cells for 30 min at 37 °C. The cells were rinsed once with cold PBS and lysed in a Tris-buffered saline containing 1% CHAPS, 20 mM β-glycerophosphate, 1 mM Na₃VO₄, 5 mM NaF, 100 ng/ml microcystin-LR, and protease inhibitor cocktail. Cell lysates were clarified by centrifugation and subjected to immunoprecipitation as follows: Cell lysates (1 mg total protein for each) were precleared with 10 µl of protein G-conjugated Dynabeads (Invitrogen) for 1 h with rotation. The cleared lysates were incubated with 5 µg of PS/2 antibody (MilliporeSigma #CBL1304) for 12 h, and then mixed with 25 µl of protein G-conjugated Dynabeads for 3 h with rotation. The beads were washed three times with 1 ml of cold PBS containing protease inhibitor cocktail. The final protein precipitates were analyzing by SDS-PAGE with mass-comparable silver staining. Protein bands were excised and subjected to the mass spectrometry for peptide sequencing.

## Mass spectrometry

Silver-stained protein bands were excised from gels with a scalpel, destained with 15 mM K₄FeCN₆ /50 mM sodium thiosulfate and washed to remove destaining reagent. The pH of samples was adjusted to 8.0 by the addition of 200 mM NH₄HCO₃. Gels were dehydrated with acetonitrile, rehydrated with 25 mM NH₄HCO₃ containing 20 ng of sequencing grade trypsin (Promega Co.) and incubated at 37 °C for 15–17 h. Peptides were then extracted with 30 µL of 60% acetonitrile/0.1% TFA and evaporated to dryness for MS analysis. Formic acid (final concentration 1%) was added to facilitate electrospray. MS/MS analysis for peptide sequencing was performed by nano flow reversed-phased HPLC/ESI/MS with a mass spectrometer (Q-TOF Ultima™ global, Waters Co. UK). Tryptic digests (5 µl) were dissolved in water/ACN/formic acid; 95:5:0.2, v/v and injected onto a C₁₈ reversed-phase 75 µm i. d. × 150 mm analytical column (3 µm particle size, Atlantis™ dC18, Waters) with an integrated electrospray ionization Silica-Tip™ (± 10 µm, New Objective, USA). Peptides were desalted prior to separation, using a trap column (i.d. 0.35 × 50 mm, OPTI-PAK™ C₁₈, Waters) cartridge and were eluted with a linear gradient of 5–80% water/ACN/formic acid; 5:95:0.2, v/v over 120 min. Chromatography was performed using the instrument's control software MassLinx of Q-TOF Ultima™ global (Version 4.0, Waters Co. U.K.). The mass spectrometer recorded scan cycles composed of one MS scan followed by MS/MS scans of the three most abundant ions in each MS scan within 10 s window. Acquired spectra were processed using Turbo SEQUEST software (version 2.0, Thermo Finnigan, San Jose, CA) and matched against amino acid sequences in mouse SwissProt database (version 57.8) or mouse NCBI database (version 2.4) using Mascot search (version 2.2.06, Matrix Science, London, UK) and MODi (Korea, http://modi.uos.ac.kr/modi/). Search parameters allowed 0.3 Da tolerance for peptide and fragment ions; up to one missed cleavage allowed, acetylation (K), deamidation (N,Q), methylation (K), dimethylation (K), pyroglu (N-term E, Q), oxidation (M), phosphorylation (S,T,Y), ubiquitination (K), propionamidation, or cabamidomethylation (C) were set as a variable modification but not fixed modification. Results were scored using probability-based Mowse score (Protein score is $-10 \times \log(p)$ where $p$ is the probability that the observed match is a random event. A protein score greater than 50 was determined to be significant ($p < 0.05$). A minimum total score of 50, comprising at least a peptide match of ion score more than 20, was arbitrarily set as threshold for acceptance. All reported assignments were verified by automatic and manual interpretation of spectra.

## Immunoblotting

Twenty-five milligrams CD11b+ bone-marrow-derived myeloid cell or splenic T cell lysates were prepared as described for immunoprecipitation were electrophoresed on 10% SDS-PAGE gels and immunoblotted to detect MLYK210 (MYLK N-17, #sc-22223 or H195 Santa Cruz Biotechnology, #sc25428 each at 1:1000 dilution), integrin α4 (EPR1355Y, Abcam #ab81280 at 1:1000 dilution), HSP90 (Santa Cruz sc101494 at 1:1000 dilution), pSer18/pThr19 myosin light chain (Cell Signaling 3674 at 1:1000 dilution), myosin light chain 2 (Cell Signaling 3672 at 1:1000 dilution), pAkt (244F9, Cell Signaling at 1:1000 dilution), Akt (11E7, Cell Signaling at 1:1000 dilution), Flag tag (F7425, Sigma at 1:1000 dilution), myosin heavy chains A (Cell Signaling 3403 at 1:1000 dilution), myosin heavy chain B (Cell Signaling 3404 at 1:1000 dilution), myosin heavy chains C (Cell

Signaling 3405 at 1:1000 dilution), paxillin (Santa Cruz 5574 at 1:1000 dilution), or Talin (Santa Cruz H-300 at 1:1000 dilution). All antibodies were diluted in TBST+ 5% BSA. Images of uncropped blots with molecular weight markers are included in the Source Data, available at https://doi.org/10.6084/m9.figshare.19358147.v1 and in the Source Data file that accompanies this article.

**In vivo myeloid cell trafficking studies.** $5 \times 10^6$ CFDA-labelled CD11b+ myeloid cells purified by gradient centrifugation from male and female 6–8-week-old C57BL/6 WT and $Mlck210^{-/-}$ mice in the C57BL/6 background or $5 \times 10^6$ CFDA-labelled $Mlck$ and non-silencing siRNA-transfected CD11b+ cells were injected intravenously into WT mice bearing subcutaneous d14 LLC tumors. Animals were euthanized 2 or 24 h after myeloid cell injection. Fluorescent cells accumulating in tumors and spleens were quantified at 2 h and 24 h after inoculation by flow cytometry of single-cell preparations of tumors. To quantify inoculated CDFA+ myeloid cells in tissues, tumors were excised, minced, and digested to single-cell suspensions for 1 h at 37 °C in 5 ml of Hanks Balanced Salt Solution (HBSS, GIBCO) containing 1 mg/ml Collagenase type IV, 10 µg/ml Hyaluronidase type V and 20 units/ml DNase type IV (Sigma). Red blood cells were solubilized with RBC Lysis Buffer (ThermoFisher). Single-cell suspensions ($10^6$ cells in 100 µl total volume) were incubated with 0.5 µg/ml propidium iodide and then with Fc-blocking reagent (Becton Dickinson) or serum. Cells were gated on SSC-A vs FSC-A to identify cells, then gated on PI negative populations to identify live cells and then on FITC vs SSC-A to identify and quantify green-fluorescent myeloid cells. Facs analysis was performed using FloJo software (Treestar, Inc).

**Tumor studies.** C57BL/6 Lewis lung carcinoma (LLC), LSL-Kras$^{G12D/+}$;LSL-p53$^{R172H/+}$;Pdx1Cre pancreatic carcinoma (LMP), or HPV+ MEER head and neck squamous cell carcinoma cells were cultured in antibiotic- and fungicide-free DMEM media containing 10% serum; cells tested negative for mycoplasma. $5 \times 10^5$ LLC or LMP cells or $3 \times 10^6$ HPV+ tumor cells were injected subcutaneously into syngeneic 6-to 8-week-old wildtype C57BL6 or $Mlck210^{-/-}$ mice in the C57BL6 background ($n = 8$–10). For LLC and LMP studies, female mice were used. For HPV+ Meer studies, male mice were used. Tumor dimensions were recorded at regular intervals, typically every 3 days. Tumors were excised at 15–48 days. Final tumor volumes and/or weights were obtained, and tumors were digested for flow cytometry or cryopreserved in O.C.T., cryosectioned into 5 µm and stained with hematoxylin and eosin or specific antibodies. LLC tumors were analyzed for areas of necrosis using Metamorph Microscopy Automation and Image Analysis (Molecular Devices, San Jose, CA). In some experiments, animals were normalized into two groups seven days after tumor cell inoculation, each with a starting volume of 10 mm$^3$. Animals were then treated with daily i.p. injections of either 0.1 mg **MW01-022AZ** in 1% DMSO or vehicle (0.1% DMSO) control.

In other experiments, wildtype female C57BL6 or $Mlck210^{-/-}$ mice in the C57BL6 background bearing LLC tumors ($n = 6$–8) were treated with daily i.p. injections of 1 mg clodronate or control liposomes (Liposoma Research Liposomes # CP-005-005) in 200 µl on day 7,11 and 15 after tumor inoculation. Tumors dimensions were recorded at regular intervals, typically every 1–2 days. Tumors were excised at 18 days after implantation and tumors, spleens and livers were excised for further analysis by flow cytometry. Alternatively, WT and $Mlck210^{-/-}$ C57Bl6 male mice bearing HPV+ MEER tumors ($n = 8$–11) were treated with i.p. injections of 100 µg anti-CD8 antibodies (BioXcell In Vivo Plus Clone YTS 169.4, #BE0117) or saline on days 27, 29, 32, 41, and 44 after tumor inoculation. Tumor volumes were measured every 2–3 days. Tumors and spleens were harvested on day 48 after tumor inoculation for flow cytometry analysis.

**Bone marrow transplantation and tumor studies.** Bone-marrow-derived cells were aseptically harvested from 6–8-week-old wildtype or $Mlck210^{-/-}$ female mice in the C57Bl6 background by flushing leg bones of euthanized mice with phosphate buffered saline containing 0.5% BSA and 2 mM EDTA, incubating cells in red cell lysis buffer (155 mM NH$_4$Cl, 10 mM NaHCO$_3$ and 0.1 mM EDTA) and centrifuging over Histopaque 1083. Approximately $5 \times 10^7$ BMDC were purified by gradient centrifugation from the femurs and tibias of a single mouse. Two million cells were intravenously injected into tail veins of lethally irradiated (1000 rad) 6-week-old syngeneic recipient female wildtype C57BL6 mice. After 4 weeks of recovery and confirmation of successful transplant, $5 \times 10^5$ tumor cells were subcutaneously inoculated into BM transplanted animals ($n = 8$), and tumor growth was monitored as described above. Successful engraftment of mutant bone marrow was assessed by isolating genomic DNA from peripheral blood cells of fully recovered bone-marrow-transplanted mice and performing genomic PCR.

**Immunofluorescence analysis of tumor tissue.** Sections of tumors cryopreserved in O.C.T. (5 µm) were fixed in 100% cold acetone and 3.7% paraformaldehyde, blocked with 8% normal goat serum for 2 h, and incubated with primary antibodies directed against mouse CD8 (rabbit anti-mouse CD8, ab217344, Abcam 2 µg/ml), CD31 (rat anti-mouse Pecam1, 553307, Becton Dickinson, 2 µg/ml) and F4/80 (rat anti-mouse Adgre1, BM8, ThermoFisher 2 µg/ml) for 2 h at room temperature. Sections were washed three times with PBS and incubated with extensively cross absorbed secondary antibodies coupled to Donkey anti-rabbit Alexa488 or Goat-anti-Rat Dylight 549 (Jackson Immuno, 112-506-003 2 µg/ml). Nuclei were stained

with 4′,6-diamidino-2-phenylindole (DAPI). Slides were washed and mounted in DAKO fluorescent mounting medium. Immunofluorescence images were collected on a Nikon microscope (Eclipse TE2000-U) and CD8+ cells/field, CD31+ vessels/field and F4/80 positive cells/field were quantified in 5 each ×100 fields per tumor and an average number of positive cells or vessels/field was determined for each individual tumor.

**MLCK activity assay.** Total bone marrow CD11b+ cells from WT and MLCK$^{-/-}$ mice were transfected with 2 µg of empty pcDNA3.1 plasmid, or pcDNA3.1 vector expressing FLAG-MLCK210-WT or FLAG-MLCK210-kinase-dead gene protein (provided by Dr. Daniel Watterson, Northwestern University) using Nucleofection (Lonza). Transfected cells were cultured overnight at 37 °C in RPMI+ 20% FBS+ Pen/Strep. The next day, cells were serum starved for 6 h and then stimulated with 200 ng/ml SDF-1α (PeproTech) for 15 min or left untreated. Cells were solubilized in standard RIPA buffer containing protease and phosphatase inhibitor cocktail. MLCK activity was measured by immunoblotting to detect phosphorylation of Ser18/Thr19 on myosin light chain (MLC) (Cell Signaling, #3674, diluted 1/1000 in TBST+ 5% BSA). Total MLC protein levels were detected by MLC2 antibody (Cell Signaling, #3672, 1:1/000 in TBST+ 5% BSA). Efficient expression of MLCK constructs was confirmed using anti-FLAG antibody (M2, Sigma).

**Immunofluorescence analysis of polystyrene bead-stimulated cells.** To induce integrin α4 activation on myeloid cells for immunofluorescence analysis, purified myeloid cells were incubated with polystyrene microspheres (0.9 µm diameter, # PS03005 Bangs Laboratories) that had been coated with 200 ng/ml SDF-1α or 2% BSA overnight at 4 °C. Cells were incubated with beads for 5 min at 37 °C, fixed with 1% paraformaldehyde for 15 min on ice, permeabilized for 15 min with 0.1% TX-100 on ice and immunostained with fluorochrome-conjugated anti-integrin α4 (R1/2, Biolegend 1:40 103629), anti-paxillin (H-114, Santa Cruz sc-5574, 1:50), anti-Rap1 (ThermoFisher Rap Pulldown kit, # 16120, 1:50), myosin heavy chain IA (Cell Signaling #3403, 1:50), rabbit anti-Flag (Sigma # F7425, 1:100), anti-To-pro (ThermoFisher T3605, 1:500) or Dapi (ThermoFisher 62248, 1:500). Cells were deposited on slides using a Cytospin centrifuge and analyzed by Nikon CS1 spectral confocal on a Nikon TE2000E inverted microscope using a ×100 microscope objective. Images were captured using Metamorph software.

**Isolation of single cells from murine tumors.** Tumors were isolated, minced in a petri dish on ice and then enzymatically dissociated in Hanks Balanced Salt Solution with Ca2+ and Mg2+ containing 0.5 mg/ml Collagenase IV (Sigma), 0.1 mg/ml Hyaluronidase V(Sigma), 0.6 U/ml Dispase II(Roche), 0.005 MU/ml DNAse I(Sigma), and 0.2 mg/ml soybean trypsin inhibitor(Worthington Biochemical) at 37 °C for 15 min. Red blood cells were lysed with red blood cell lysis solution, and the resulting suspension was filtered through a 70 um cell strainer to produce a single-cell suspension. Cells were then washed one time with PBS prior to incubation with fluorescently conjugated antibodies for flow cytometry.

**Flow cytometry staining and analysis.** Single-cell suspensions of dissociated tumor cells ($10^6$ cells in 100 µL total volume) were incubated first with Propidium Iodide and then with Fc-blocking reagent (Becton Dickinson) followed by fluorescently labeled antibodies and incubated at 4 °C for 1 h. Primary antibodies used for cell surface marker staining included anti-CD11b (M1/70 APC, ThermoFisher 1.25 µg/ml) and anti-Gr1 (RB6-8C5 FITC, ThermoFisher 1.25 µg/ml). Three color FACS Analysis was performed on a Becton Dickinson 4 color FACSCalibur color analyzer (Moores Cancer Center Flow Cytometry Core). Data analysis was performed using the flow cytometry analysis program FlowJo (Becton DIckinson). Cells were gated as shown in Supplementary Fig. 6b. Alternatively, single-cell suspensions of dissociated tissues ($10^6$ cells in 100 µL total volume) were incubated first with BUV395 fixable dead cell stain and then with Fc-blocking reagent (Becton Dickinson) followed by fluorescently labeled antibodies and incubated at 4 °C for 1 h. Primary antibodies used for cell surface marker staining included anti-CD45 (AF700 clone 30-F11, Biolegend #103127 1.2 µg/ml) anti-CD11b (M1/70 AF288 ThermoFisher # 14-0112-82 0.625 µg/ml) and anti-Ly6C (BV421 clone HK1.4 Biolegend #28031 1.25 µg/ml), Anti-Ly6G (Pe-Cy7 clone 1A8 BD Bioscience #560601 1.25 µg/ml), Anti-F4/80 (SBV670 clone Cl: A3-1 Bio-Rad #MCA497SBV670 1.25 µg/ml), anti-CD3 (PE clone 145-2C11 ThermoFisher #12-0031-82 2.5 µg/ml), anti-CD4 (BV650 BD Bioscience RM4-5 #63747 2.4 µg/ml), or anti-CD8 (BV785 clone 53–6.7 Biolegend #100749 2.5 µg/ml). Multi-color FACS Analysis was performed on a 5 laser Becton Dickinson LSRFortessaX20 Facs analyzer using BD Facs Diva software at the Sanford Consortium for Stem Cell Research Flow Cytometry Shared Resource. All data analysis was performed using the flow cytometry analysis program FlowJo (Becton Dickinson). Cells were gated as shown in Supplementary Fig. 6a, c.

**Software.** Turbo SEQUEST software (version 2.0, Thermo Finnigan, San Jose, CA), Mascot search (version 2.2.06, Matrix Science, London, UK), and MODi (Korea, http://modi.uos.ac.kr/modi/) were used in tandem mass spectroscopy data analysis. Prism v9 (GraphPad, San Diego, CA) was used for graphing and statistical analysis. Metamorph Microscopy Automation and Image Analysis (Molecular Devices,San Jose, CA) was used for microscopy image capture and analysis.

### Determination of sample size

*In vivo studies.* A sample size of 10 mice/group provides 80% power to detect mean difference of 2.25 standard deviation (SD) between two groups (based on a two-sample *t*-test with two-sided 5% significance level). Typically, tumor studies were performed with 8–10 or more mice per group. *In vitro studies:* A sample size of 3–5 replicates/sample for in vitro studies provided sufficient power to detect mean difference of 1–2 standard deviations (SD) between two groups (based on a two-sample *t*-test with two-sided 5% significance level).

### Statistics

All significance testing was performed in Prism by GraphPad software by unpaired two-sample Student's *t*-test for two groups with normally distributed data and by Mann–Whitney rank test for two groups when data were not normally distributed. Multiple group analyses were analyzed by one-way Anova with Tukey's posthoc testing for multiple pairwise testing (multiple groups) when data were normally distributed and a Wilcoxon rank sum test (two groups) when data were not. All mouse studies were randomized and blinded; assignment of mice to treatment groups, tumor measurement, and tumor analysis was performed by coding mice with randomly assigned mouse number, with the key unknown to operators until experiments were completed. All experiments were performed two or more times with similar results. For in vitro studies, random assignment of aliquots (cells, proteins) to groups was performed with coding of samples so that sample identity was not apparent to user during sample processing. For Facs analysis, all data were comprised of 3–5 biological replicates. Three to five biological replicates were used for RT-PCR. All studies were performed a minimum of 2–3 times.

**Reporting summary**. Further information on research design is available in the Nature Research Reporting Summary linked to this article.

## Data availability

The datasets generated and analyzed during the current study are available with this paper in the Article, Supplementary Information and Source Data file. Source data are also available in figshare at [https://doi.org/10.6084/m9.figshare.19358147.v1]. The databases SwissProt (https://www.expasy.org/resources/uniprotkb-swiss-prot) and mouse NCBI nr (https://www.ncbi.nlm.nih.gov/gene) were utilized in this study. Source data are provided with this paper.

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

## Acknowledgements

These studies were supported by NIH grants 5R01CA226909 J.A.V., 5R01CA167426 J.A.V., and 5R01DE027325 J.A.V. This project was partially supported by NCI P30CA23100 to the UCSD Moores Comprehensive Cancer Center for Flow Cytometry Shared Resource services.

## Author contributions

Immunoprecipitations, silver staining, and proteomics were performed by S.W.K. and M.C.S. Immunocytofluorescence studies were performed by M.C.S., M.P., A.D., and S.C. Flow cytometry was performed by M.C.S., M.M.K., and H.C. Cell adhesion, integrin activation, cell transfections, and immunoblotting studies were performed by M.C.S. and by H.C. In vivo experiments were performed by M.C.S., H.C., M.P., and A.G. *Mlck210*⁻/⁻ mice, **MW01-22AZ**, and Flag-tagged *Mlck210* expression constructs were provided by D.M.W.

## Competing interests

The authors declare no competing interests.
