## [Peer Review File · Nature Communications]

Reviewers' Comments:

Reviewer #1:

Remarks to the Author:

In this study, the authors decipher a molecular mechanism leading to activation of $\alpha 4\beta 1$ integrin in myeloid cells. They reveal the interaction of the 210 kDa myosin light chain kinase (MLCK210) with integrin upon cytokine stimulation. Genetic or pharmacological inhibition of MLCK210 abrogates cytokine-induced activation of Rap1 GTPases and $\alpha 4\beta 1$ integrin. They further provide evidences that MLCK210 phosphorylates and activates myosin leading to integrin activation. Inflammation and tumor growth were substantially impaired in MLCK210 knockout mice and in mice transplanted with MLCK210 knockout bone marrow. This study extends previous work from the Varner lab revealing that inflammatory stimuli promote $\alpha 4\beta 1$ integrin activation in myeloid cells through a signaling cascade that involves PI3K γ and Rap1, and subsequent recruitment of monocytes into mouse tumors. Remarkably, this work positions MLCK210 as an important player in the sequence of signaling events leading to $\alpha 4\beta 1$ integrin activation in myeloid cells.

The manuscript is very well written, with convincing and well-presented data. Overall, the methodology is logical and well defined. Although it makes an interesting and significant contribution, the following issues should be addressed:

- 1) The authors show that genetic deletion or pharmacological inhibition of MLCK210 in myeloid cells suppresses their recruitment into tumors. The contribution of MLCK210 in activating Rap1 and subsequently integrins is well demonstrated through a combination of in vitro experiments in the present study. However, there is no evidence that the effects on the recruitment of the myeloid cells into tumors and the impact on tumor growth and spread are due to a defect in $\alpha 4\beta 1$ integrin activation caused by the loss of MLCK210 protein and activity. The authors are encouraged to either provide supportive data or modify their conclusion. Furthermore, it would be interesting to show whether inhibition or silencing of MLCK210 promotes T cell recruitment and activation. This information would strengthen the conclusion proposed by the authors that MLCK210 inhibitors could be considered as potential immune oncology therapeutics.
- 2) They nicely demonstrate that MLCK210 participates in activation of $\alpha 4\beta 1$ integrin. Is that also true for other integrins? The title and the abstract of the manuscript should be more specific in regard to the integrin heterodimers regulated by MLCK210. Although it is clear that MLCK210 controls Rap1 GTPases, the contribution of paxillin is less evident. It would be of interest to decipher whether paxillin is involved as it could be specific to $\alpha 4\beta 1$ integrin.
- 3) $\alpha 4\beta 1$ integrin interacts with MLCK210. Is that a direct interaction?
- 4) Why using siRNA-transfected murine myeloid cells to knockdown the expression of MLCK210 when the authors make skillful use of MLCK210 knockout myeloid cells?
- 5) In figure 1d, the authors should include immunoblots of total homogenates before immunoprecipitation. This valuable information would help appreciate the efficiency of the immunoprecipitation and the amplitude of the interaction.
- 6) Showing the localization of endogenous MLCK210, but not overexpressed protein, in myeloid cells would be helpful. In addition, I do not think that the immunofluorescence data presented throughout the manuscript are supportive of the statement that MLCK210 is required for integrin clustering in response to cytokine stimulation and I do not share the same conclusion with the authors. My interpretation from these experiments is that MLCK210 colocalizes with the integrin and their distribution, but not clustering, is changed upon cytokine stimulation.
- 7) Experiments aiming at measuring $\alpha 4\beta 1$ integrin activation with soluble ligand binding or using an activation-specific antibody should be controlled for the total expression of the integrin at the surface level. Addition of manganese partially addresses this point, but it also reveals that siRNA-mediated silencing of MLCK210 may reduce the surface expression of $\alpha 4\beta 1$ integrin according to data presented in Figure 2e.

- 8) Scale bars are missing in all immunofluorescence panels.
- 9) Figure 5e is not cited.
- 10) There is no evidence that siRNA-mediated silencing of Myh10 reduces its expression. Please provide immunoblot or RT-qPCR data.
- 11) In Figure 5i, please include control images for MLCK210 wt and kd expression.
- 12) It is unclear what mice are used in panels m-p of Figure 6. Please specify.
- 13) Method description for migration experiments is missing.
- 14) References 6 and 31 are identical.
- 15) There are a few typos: page 4 line 8 (inflammation and progression); page 6 line 25 (should be inhibition, and not deletion, of MLCK210); page 9 line 25 (integrin $\alpha 4$ mediated ??); page 16 line 8 (were identified as gelsolin); page 16 line 11 (were immunoblotted to detect); Figure 2c x-axis (MLCK210-).
- 16) There is no consistency in the nomenclature of the MW01022AZ inhibitor.

Reviewer #2:

Remarks to the Author:

In this article Schmit et al. identify MLCK210 as an important mediator that mediate the transmission of the GPCR signaling to the integrin $\alpha 4\beta 1$ and mediating myeloid cell adhesion and chemotaxis. Despite some important antitumor effects (Fig.6) and the use of adequate strategy and tools to dissect the above mentioned pathways, the absence of key controls, a poor graphic representation of the data and the lack of adequate characterization of the tumor microenvironment drastically reduce the enthusiasm for this manuscript. Please find below my concerns.

Major concerns

1. Figure 1b, it is unlikely that only 2 proteins were identified by mass spectrometry. A table with the different proteins identified, relative scores and peptides should be provided at least in the supplementary material. Additionally, immunoprecipitation and mass spec should be repeated at least twice and data from both experiments provided.
2. The authors identified as one of the binders if the integrin $\alpha 4\beta 1$ MLTK210 it is unclear how this isoform was differentiated by the many other ones existing for this gene. The experiment with MLTK210^{-/-} may only exclude the MLTK108.
3. As presented fig. 1e,g, fig.5i and suppl.fig.3a,b are anecdotal. In addition to the images provided as example a numeric analysis on multiple cells from multiple experiments should be provided.
4. Given the heterogeneity of myeloid cells, additional markers such for example Ly6c, Ly6g, F4/80 should be used to differentiate the myeloid subsets.
5. Data should be better presented to understand the variation between groups and controls should be better specified in the legends. For example:
 - a. Bar plot should be changed in boxplot
 - b. SEM should be avoided for the use of Standard Deviation or 50, 75, 90 percentile for box plot
 - c. it is unclear what the authors mean with nonsil or control transfected in fig2b is this scrambled siRNA?
6. Figure 3a-c Vehicle control or better drug inactive enantiomer should be included.
7. Figure 3g,h actual p value should be reported, sem avoided, and data should be expressed as dot plot.
8. Figure 5b, c, d, g scrambled controls are missing
9. Figure 5d,f, h please use SD or dot blot
10. Figure 6k,d a better phenotypic analysis of tumor microenvironment should be provided. It is puzzling that the strong anti-tumor effect observed in figure 6c-h is mediated by a modest

reduction of putative monocytic MDCS Gr1^{low} and no reduction in PMN-MDSC and Gr1⁻ (macrophage?) myeloid cells. The phenotypic characterization should include myeloid and lymphoid subsets. Additionally, since all the immune system in these experiments is impaired for MLCK210, how can the author exclude the involvement of other population?

Minor concerns

Reference 16 is wrong. It should be change with <https://doi.org/10.1073/pnas.1031595100>

Reviewer #3:

Remarks to the Author:

This study by Schmid et al explores myeloid alpha4beta1 integrin inside-out signaling, elucidating a new pathway involving PI3Kgamma, MLCK210 and Rap1 that functions in integrin inside-out signaling that regulates myeloid cell adhesion. At the same time, MLCK210 in the tumor (immune cell) microenvironment is required for tumor growth in syngeneic mouse tumour models. The question addressed is an interesting one that is of interest to the wider scientific community. However I have a number of reservations.

Major points.

1. The authors examine CD11b-positive human donor blood or mouse bone marrow-derived myeloid cells. While it is commendable that they used primary cells, this is a heterogeneous cells population, which is comprised primarily of neutrophils and monocytes (as well as others), with monocytes again being a heterogeneous group of cells comprised of subsets with dramatically different functions. These different cell types are characterised by differential receptor expression and do not share all signaling pathways. Moreover, signaling in human and mouse immune cells is frequently not conserved. The origin of the human donor blood is described as 'outdated samples' from a blood bank. Given the short lifespan of neutrophils and classical monocytes in the circulation (~24hrs), the majority of these cells are likely to be apoptotic or at least ready to home back to the bone marrow, but neither apoptotic cells nor upregulation of markers characteristic of aged cells seem to have been performed. The cells actually used in experiments should be clearly identified according to a range of cell surface markers. Mouse bone marrow contains a large reservoir of largely immature monocytes and of neutrophils (mostly band cells), which again may result in different behaviour to that found with circulating cells and tissue cells, where monocytes in particular are prone to undergoing many changes in terms of surface markers and pro- versus anti-inflammatory functions.
2. The authors examine alpha4beta1 integrin, a major integrin regulating extravasation in T cells. In myeloid cells, however, alpha4beta1 is a relatively minor integrin while beta2 integrins (LFA1, Mac1) are more important. Does the MLCK210-containing pathway also regulate these beta2 integrins? Does the pathway function in T cells? The authors use a number of different substrates to address this, but integrins are notoriously promiscuous in terms of ligand binding, meaning this is not sufficient. Similarly the authors do not show that alpha4beta1 integrin is being activated, since the only direct assay employed uses an activation epitope-specific anti-beta1 antibody, that will inform on the activation status of any beta1 containing integrins.
3. Quantitative analysis of the integrin activity status (HUTS21 binding; VCAM-1 binding with suspension cells) should be performed instead of (or in addition) to the functional assay (cell adhesion) wherever a new player is being analyzed e.g. Figs 4C,G, 5F
4. The major assay employed in an analysis of cell adhesion under static conditions with cells adhering to VCAM-1 coated plastic or HUVECs. Here myeloid cells are calcein labelled and adhesion is quantified using a plate reader. A better way to analyse integrin function in the context of static adhesion would be an analysis of cell adhesion microscopically analyzing cell number and cell size (in pixels), to inform on adhesion as well as spreading, both of which require integrin activity.
5. Relatedly, the authors co-stimulate the cells with chemokine / cytokine to induce integrin activation. Stimuli employed are IL-1beta, IL-6, SDF-1alpha, tumour cell medium, TNF and VEGF-A. Intriguingly, all these stimuli induce a very similar degree of activation in the adhesion experiments, suggesting that the assay has not been carefully calibrated and the linear range of the assay has been exceeded. Most importantly, it is unclear how some of the stimuli used can

cause a significant activation, given that the cells analysed are likely predominantly neutrophils, with a smaller degree of monocytes/macrophages. Circulating neutrophils do not express VEGF-R (although there has been a description of a subtype of cancer neutrophils that do, Massena et al 2015 Blood 125, 2016), and neither do the vast majority of circulating monocytes (Nowall et al 2004 Circulation 110, 2699). Similarly, IL-1beta does not stimulate isolated neutrophils (Prince et al 2004 Am J Pathol 165, 1819). This demonstrates again the importance of carefully addressing point 1.

6. In addition to static adhesion assays, leukocyte adhesion should be analysed under more physiological flow conditions in a parallel plate chamber.

7. Figure 2F shows that MLCK210 knock-down cells have a migration defect. This is a very important observation, given that recruitment to tumors depends upon migration. Is this a transwell assay? The authors should perform time-lapse imaging on different substrates to identify whether the defect is due to an adhesion or a de-adhesion defect; fantastic reagents (e.g. function blocking antibodies) exist for human integrins that would allow the identification of the integrin required. Transendothelial migration should also be investigated.

8. To measure integrin clustering and colocalization of proteins involved in the pathway, the authors stimulate the cells with BSA- or CXCL12-coated beads. The beads are described as very large (9um) and should be shown in the first instance used images – presumably the cells are undergoing frustrated phagocytosis? The analysis of colocalization is not convincing. Rather than co-localizing the integrin appears to be localizing to the same area of the cell but further to the plasma membrane than MLCK210 (Fig 1e; S3A), Rap (Fig S3A) and Myosin (Fig 5e) giving the impression of sequential rather than co-localized stain. At best there may be instances of partial co-localization. Use of a plasma membrane dye may be helpful here. Moreover single representative cells are not sufficient to draw firm conclusions and analysis of a larger number of images should be presented. Please also note that integrin clustering is a molecular event that occurs on a scale too small to be visualized with an ordinary light microscope, so this term should be avoided, or else superresolution microscopy or immuno-EM should be employed to analyze molecular events.

9. Fig 6A,B show recruitment of adoptively transferred control or MLCK210-deficient myeloid cells to tumors or the spleen. It is unclear why the authors analyze recruitment to the spleen rather than the bone marrow and/or the lung where adoptively transferred cells tend to marginate; also blood cells should be shown. How many of the adoptively transferred cells do remain viable after 24 hours?

10. The tumour data, especially those performed with bone marrow chimeras are very interesting, though do not appear overly connected to the work presented in the first 5 figures. Importantly the experiments presented do not show that tumour growth is dependent upon MLCK10 in myeloid cells – it could be due to any radiosensitive cell type(s). To test the dependency on a particular immune cell type depletion experiments ought to be performed. In addition, experiments in which mice with established tumours are treated with the MW01 compound should be performed to test whether blocking MLCK210 is sufficient to reverse tumour growth.

Minor points.

This manuscript appears to have been put together rather hastily.

1. The methods are too brief for readers to repeat any of the experiments. Incubation/stimulation times, agonist concentrations, objectives used for microscopy details on image analysis etc are all missing.

2. The figure legends do not put experiments into context and should be expanded. What is the migration assay shown in Fig 2F (this is also not mentioned in the methods section)? Definitions are also missing (e.g. TOPRO, Fig S3C). There are no details on statistical analysis performed, with any of in vitro data shown.

3. It should be made clear throughout whether experiments were performed with human or mouse cells by clearly labelling the figures. Key experiments should be performed with both, since signaling is often not conserved between the two species.

4. Multiple Western blot-based experiments rely on a single representative example. Several experiments ought to be performed in each instance, signal quantification should be performed and shown. This is especially critical where Western blots are performed to inform on the activity status of a protein of interest.

5. All micrographs are missing scale bars.

6. A section that informs on the statistical analysis of in vitro experiments is missing.

7. Individual points ought to be presented rather than bar graphs, especially in Fig 6 but ideally throughout.
8. The timing of the data shown in Fig 6O,P should be indicated in the legend.

Response to Reviewers

Reviewer #1

In this study, the authors decipher a molecular mechanism leading to activation of $\alpha 4\beta 1$ integrin in myeloid cells. They reveal the interaction of the 210 KDa myosin light chain kinase (MLCK210) with integrin upon cytokine stimulation. Genetic or pharmacological inhibition of MLCK210 abrogates cytokine-induced activation of Rap1 GTPases and $\alpha 4\beta 1$ integrin. They further provide evidence that MLCK210 phosphorylates and activates myosin leading to integrin activation. Inflammation and tumor growth were substantially impaired in MLCK210 knockout mice and in mice transplanted with MLCK210 knockout bone marrow. This study extends previous work from the Varner lab revealing that inflammatory stimuli promote $\alpha 4\beta 1$ integrin activation in myeloid cells through a signaling cascade that involves PI3K γ and Rap1, and subsequent recruitment of monocytes into mouse tumors. Remarkably, this work positions MLCK210 as an important player in the sequence of signaling events leading to $\alpha 4\beta 1$ integrin activation in myeloid cells.

The manuscript is very well written, with convincing and well-presented data. Overall, the methodology is logical and well defined.

We greatly appreciate the reviewer's positive comments.

Although it makes an interesting and significant contribution, the following issues should be addressed:

1) The authors show that genetic deletion or pharmacological inhibition of MLCK210 in myeloid cells suppresses their recruitment into tumors. The contribution of MLCK210 in activating Rap1 and subsequently integrins is well demonstrated through a combination of in vitro experiments in the present study. However, there is no evidence that the effects on the recruitment of the myeloid cells into tumors and the impact on tumor growth and spread are due to a defect in $\alpha 4\beta 1$ integrin activation caused by the loss of MLCK210 protein and activity. The authors are encouraged to either provide supportive data or modify their conclusion.

We have previously published extensively that integrin \$\alpha 4\beta 1\$ is required for myeloid cell trafficking to tumors and for tumor growth (Schmid, M.C., et al. Combined blockade of integrin- \$\alpha 4\beta 1\$ plus cytokines SDF-1 \$\alpha\$ or IL-1 \$\beta\$ potently inhibits tumor inflammation and growth. *Cancer Res* **71**, 6965-75 (2011). We furthermore showed that PI3K \$\gamma\$ activates \$\alpha 4\beta 1\$ to promote trafficking to tumors and tumor growth (Schmid, M.C., et al. Receptor tyrosine kinases and TLR/IL1Rs unexpectedly activate myeloid cell PI3K \$\gamma\$, a single convergent point promoting tumor inflammation and progression. *Cancer Cell* **19**, 715-27 (2011). We then showed that Rap1 is required for PI3K \$\gamma\$ mediated integrin activation, myeloid cell trafficking and tumor growth Schmid, M.C., et al. PI3-kinase \$\gamma\$ promotes Rap1a-mediated activation of myeloid cell integrin \$\alpha 4\beta 1\$, leading to tumor inflammation and growth. *PLoS One* **8**, e60226 (2013). Finally, in this manuscript, we show that MLCK210 is required for PI3K \$\gamma\$ dependent \$\alpha 4\beta 1\$ integrin activation and adhesion and for myeloid cell trafficking and tumor growth. Taken together, our new studies show that inhibition of PI3K \$\gamma\$, MLCK210, Myosin and Rap1 suppress the same integrin \$\alpha 4\beta 1\$ activation pathway that promotes myeloid cell trafficking to tumors and tumor growth. We have modified the introduction, results and discussion sections to highlight that this work provides data supporting this pathway.

Furthermore, it would be interesting to show whether inhibition or silencing of MLCK210 promotes T cell recruitment and activation. This information would strengthen the conclusion proposed by

the authors that MLCK210 inhibitors could be considered as potential immune oncology therapeutics.

Thank you for the suggestion. We now show in Figure 7 and 8 that MLCK210 deletion and inhibition suppresses angiogenesis and stimulates T cell recruitment to tumors.

2) They nicely demonstrate that MLCK210 participates in activation of $\alpha 4\beta 1$ integrin. Is that also true for other integrins? The title and the abstract of the manuscript should be more specific in regard to the integrin heterodimers regulated by MLCK210.

Our studies show that only integrin $\alpha 4\beta 1$ and MLCK210 regulate adhesion to endothelium and trafficking to tumors in response to tumor derived cytokines. We have modified the title and abstract as well as the text to more specifically refer to MLCK210 activation of only $\alpha 4\beta 1$.

Although it is clear that MLCK210 controls Rap1 GTPases, the contribution of paxillin is less evident. It would be of interest to decipher whether paxillin is involved as it could be specific to $\alpha 4\beta 1$ integrin.

Previous studies have shown that paxillin binding to integrin $\alpha 4\beta 1$ facilitates integrin activation in as $\alpha 4Y991A$ mutation and the PI3K γ deletion in myeloid cells prevent paxillin binding to integrin $\alpha 4\beta 1$ as well as integrin activation, trafficking and tumor growth. We did not have an opportunity to directly explore a functional role for paxillin downstream of MLCK210 but agree that would be an excellent next step.

3) $\alpha 4\beta 1$ integrin interacts with MLCK210. Is that a direct interaction?

As MLCK210 activates Rap1 and we previously showed that RapGEFs CalDAG-GEF1 and CalDAG-GEF2 are required for Rap activation and integrin $\alpha 4\beta 1$ mediated adhesion [Schmid, M.C., et al. PI3-kinase γ promotes Rap1a-mediated activation of myeloid cell integrin $\alpha 4\beta 1$, leading to tumor inflammation and growth. *PLoS One* 8, e60226 (2013)] it is likely the interaction is as part of a complex of adaptors and kinases that nucleate around PI3K γ and integrin $\alpha 4\beta 1$. This would be an excellent question to explore in a follow up manuscript.

4) Why using siRNA-transfected murine myeloid cells to knockdown the expression of MLCK210 when the authors make skillful use of MLCK210 knockout myeloid cells?

To explore whether endogenous MLCK mediates integrin $\alpha 4\beta 1$ clustering in response to cytokine stimulation as suggested by studies with transfected knockout cells we depleted WT cells of MLCK 210 by siRNA knockdown.

5) In figure 1d, the authors should include immunoblots of total homogenates before immunoprecipitation. This valuable information would help appreciate the efficiency of the immunoprecipitation and the amplitude of the interaction.

We apologize for the mislabeling of Figure 1d. This blot is an immunoblot of plasma membranes purified from basal or SDF-stimulated myeloid cells that were probed for the presence of MLCK210. These data show that MLCK210 associates with the plasma membrane only upon chemokine stimulation, while integrin $\alpha 4\beta 1$ is constitutively present in plasma membranes. Taken together with the co-immunoprecipitation data from Figure 1a and the co-localization data in

Figure 1e, these results show that cytokine stimulation induces MLCK210 to associate closely with integrin $\alpha 4$ in an MLCK210 kinase dependent manner.

6) Showing the localization of endogenous MLCK210, but not overexpressed protein, in myeloid cells would be helpful. In addition, I do not think that the immunofluorescence data presented throughout the manuscript are supportive of the statement that MLCK210 is required for integrin clustering in response to cytokine stimulation and I do not share the same conclusion with the authors. My interpretation from these experiments is that MLCK210 colocalizes with the integrin and their distribution, but not clustering, is changed upon cytokine stimulation.

We were unable to find an anti-MLCK antibody that would recognize ONLY the 210 kDa isoform and analyzing the endogenous MLCK210 protein by immunofluorescence is not possible at this point in time for that reason. The revised data in figure 1e, 1g and Supplementary Figure 1b show that MLCK210 kinase expression and activity are required to permit integrin clustering at the sites of bead binding.

7) Experiments aiming at measuring $\alpha 4\beta 1$ integrin activation with soluble ligand binding or using an activation-specific antibody should be controlled for the total expression of the integrin at the surface level. Addition of manganese partially addresses this point, but it also reveals that siRNA-mediated silencing of MLCK210 may reduce the surface expression of $\alpha 4\beta 1$ integrin according to data presented in Figure 2e.

We have now included the cell counts per peak in each FACs profile for the ligand binding assays, which show that siRNA-mediated silencing of MLCK210, MLCK210 inhibitors, and MLCK210 deletion do not inhibit the number of cells per peak. As you discussed, the Mn^{2+} control shows that loss of MLCK210 function and expression does not affect total inducible $\alpha 4\beta 1$ binding of VCAM, a selective ligand for integrin $\alpha 4$.

8) Scale bars are missing in all immunofluorescence panels.

Scale bars are now included.

9) Figure 5e is not cited.

This figure is now cited.

10) There is no evidence that siRNA-mediated silencing of Myh10 reduces its expression. Please provide immunoblot or RT-qPCR data.

Myh10 is not expressed at significant levels in myeloid cells. This siRNA was included as a non-silencing control.

11) In Figure 5i, please include control images for MLCK210 WT and KD expression.

These are provided in Supplementary Figure 3c.

12) It is unclear what mice are used in panels m-p of Figure 6. Please specify.

These were WT C57Bl6/J mice. This has been added to the legend of new Figure 8, which describes the in vivo drug studies.

13) Method description for migration experiments is missing.

This has been added as a new section in the Methods section.

14) References 6 and 31 are identical.

Reference 31 has been removed.

15) There are a few typos: page 4 line 8 (inflammation and progression); page 6 line 25 (should be inhibition, and not deletion, of MLCK210); page 9 line 25 (integrin α 4 mediated ??); page 16 line 8 (were identified as gelsolin); page 16 line 11 (were immunoblotted to detect); Figure 2c x-axis (MLCK210-).

We have corrected these errors.

16) There is no consistency in the nomenclature of the MW01022AZ inhibitor.

Thank you for pointing this out. We have now corrected this reference to MW01-022AZ.

Reviewer #2 (Remarks to the Author): with expertise in myeloid cells and cancer

In this article Schmid et al. identify MLCK210 as an important mediator that mediate the transmission of the GPCR signaling to the integrin α 4 β 1 and mediating myeloid cell adhesion and chemotaxis. Despite some important antitumor effects (Fig.6) and the use of adequate strategy and tools to dissect the above-mentioned pathways, the absence of key controls, a poor graphic representation of the data and the lack of adequate characterization of the tumor microenvironment drastically reduce the enthusiasm for this manuscript. Please find below my concerns.

Thank you for the positive comments and for identifying areas for improvement. We have now addressed the important concerns you have raised.

Major concerns

1. Figure 1b, it is unlikely that only 2 proteins were identified by mass spectrometry. A table with the different proteins identified, relative scores and peptides should be provided at least in the supplementary material. Additionally, immunoprecipitation and mass spec should be repeated at least twice and data from both experiments provided.

In these experiments, our goal was to identify major proteins that associate with integrin α 4 β 1 upon stimulation by tumor derived factors. We set out to sequence only those proteins that could be directly visualized and excised from gels. Only two such proteins were identified. Each band was individually excised and sequenced. One of these bands was a 210 kDa protein with the

amino acid sequence of MLCK, properties consistent with the protein MLCK210. MLCK210 was then further studied in detail in this manuscript. We performed immunoprecipitation and silver staining analyses several times, with identical results.

2. The authors identified as one of the binders of the integrin $\alpha 4\beta 1$ as MLCK210. It is unclear how this isoform was differentiated by the many other ones existing for this gene. The experiment with MLCK210^{-/-} may only exclude the MLCK108.

We observed only two silver-stained proteins that co-precipitated with integrin $\alpha 4\beta 1$, a 210 kDa protein and an 80 kDa protein. Upon sequencing, these were identified as MLCK and gelsolin, respectively. Two representative full-length gels of these immunoprecipitations (Supplementary Figure 1a) clearly document that only a 210 kDa protein and an 80 kDa protein co-immunoprecipitated with integrin $\alpha 4\beta 1$. A search of the literature revealed that a novel 210 kDa isoform of MLCK, MLCK210, had been identified in endothelial cells, where it plays a role in vascular leak. There may be roles for other isoforms of MLCK in regulating other integrins, but our studies show this isoform is the only isoform that regulates $\alpha 4\beta 1$ activity.

3. As presented fig. 1e,g, fig.5i and suppl.fig.3a,b are anecdotal. In addition to the images provided as example a numeric analysis on multiple cells from multiple experiments should be provided.

We have now included images of several cells for each condition. The additional images are found in the Supplementary Figures 1b and 3c.

4. Given the heterogeneity of myeloid cells, additional markers such for example Ly6c, Ly6g, F4/80 should be used to differentiate the myeloid subsets.

We have extensively performed FACs analysis to detect myeloid populations in LLC tumors. The Ly6g and Ly6c markers identify the very same populations as Gr1 in our studies. The Gr1neg population is Ly6c⁻, Ly6g⁻ while the Gr1lo population is Ly6c⁺, Ly6g⁻. The Gr1hi population is Ly6clo, Ly6ghi.

5. Data should be better presented to understand the variation between groups and controls should be better specified in the legends. For example:

a. Bar plot should be changed in boxplot

We changed all graphs to dot plots or dot plots with overlaid bar graphs.

b. SEM should be avoided for the use of Standard Deviation or 50, 75, 90 percentile for box plot

We now provide all individual points as well as the original data in a single source data file. Therefore, we continue to use SEM.

c. it is unclear what the authors mean with nonsil or control transfected in fig2b is this scrambled siRNA?

We used Qiagen All Stars siRNA as a negative control for transfections. We have relabeled figures to change “nonsilencing” siRNA to All Stars siRNA.

6. Figure 3a-c Vehicle control or better drug inactive enantiomer should be included.

We always use vehicle as a control for drug studies. The figures have now been labeled appropriately.

7. Figure 3g,h actual p value should be reported, sem avoided, and data should be expressed as dot plot.

We now report actual p values in all figures and show all data as dot plots.

8. Figure 5b, c, d, g scrambled controls are missing

We now show “scrambled” negative controls (All Stars siRNA) in all graphs using siRNA except 5c, where Myh10 itself serves as a negative control, as it is not expressed in myeloid cells.

9. Figure 5d,f, h please use SD or dot blot

Dot plots are now used.

10. Figure 6k,d a better phenotypic analysis of tumor microenvironment should be provided. It is puzzling that the strong anti-tumor effect observed in figure 6c-h is mediated by a modest reduction of putative monocytic MDSC Gr1^{low} and no reduction in PMN-MDSC and Gr1⁺ (macrophage?) myeloid cells. The phenotypic characterization should include myeloid and lymphoid subsets. Additionally, since all the immune system in these experiments is impaired for MLCK210, how can the author exclude the involvement of other population?

We have now performed IHC for macrophages, blood vessels and T cells for BMT tumors and drug treated tumors (Figures 7h-l and 8g-h). We show that MLCK210 depletion reduces macrophage content of tumors, decreases blood vessel density (which is a function of decreased macrophage content) and increases T cell content, although LLC tumors are T cell poor tumors in general (Kaneda, et al 2016).

Our studies of MLCK210 deletion in bone marrow and inhibition by the pharmacologic inhibitor MW01-22AZ show identical suppression of myeloid cell recruitment to tumors but increasing T cell recruitment to tumors. As these results are identical to results observed for $\alpha 4\beta 1$, PI3K γ , and Rap1 inhibition and deletion and our studies have shown that LLC and LMP tumors studied in this manuscript are T cell ignorant tumors (studies published in *Cancer Research*, *Cancer Cell*, *PlosOne*, *Nature Communications*, *Cancer Immunology Research* and *Nature*), these results support the conclusion that MLCK210 promotes myeloid cell trafficking into tumors, where myeloid cells promote tumor growth by suppressing T cell recruitment and stimulating angiogenesis.

Minor concerns

Reference 16 is wrong. It should be change with <https://doi.org/10.1073/pnas.1031595100>

This has been corrected and the proper reference is now #15.

Reviewer #3 (Remarks to the Author): with expertise in PI3K signaling, integrins in myeloid cells

This study by Schmid et al explores myeloid $\alpha 4\beta 1$ integrin inside-out signaling, elucidating a new pathway involving PI3K γ , MLCK210 and Rap1 that functions in integrin inside-out signaling that regulates myeloid cell adhesion. At the same time, MLCK210 in the tumor (immune

cell) microenvironment is required for tumor growth in syngeneic mouse tumour models. The question addressed is an interesting one that is of interest to the wider scientific community.

We appreciate these positive comments.

Major points.

1. The authors examine CD11b-positive human donor blood or mouse bone marrow-derived myeloid cells. While it is commendable that they used primary cells, this is a heterogeneous cells population, which is comprised primarily of neutrophils and monocytes (as well as others), with monocytes again being a heterogeneous group of cells comprised of subsets with dramatically different functions. These different cell types are characterized by differential receptor expression and do not share all signaling pathways. Moreover, signaling in human and mouse immune cells is frequently not conserved.

The origin of the human donor blood is described as 'outdated samples' from a blood bank. Given the short lifespan of neutrophils and classical monocytes in the circulation (~24hrs), the majority of these cells are likely to be apoptotic or at least ready to home back to the bone marrow, but neither apoptotic cells nor upregulation of markers characteristic of aged cells seem to have been performed. The cells actually used in experiments should be clearly identified according to a range of cell surface markers.

We apologize for an error, which has been corrected, in the Methods section of the paper describing use of human blood for preparation of human myeloid cells. Myeloid cells were actually purified from fresh human peripheral blood cell preparations collected by plasmapheresis by the San Diego Blood Bank.

Gradient centrifugation methods were used to purify mononuclear cells from mouse bone marrow and human plasmapheresis samples. This procedure eliminates most polymorphonuclear granulocytes as they centrifuge to the bottom of the gradient. After CD11b bead isolation, we were left with mainly monocytes in the cell populations that we further studied. To specifically study the role of MLCK210 in monocytes and granulocytes (Figure 2c), these cells were purified by flow cytometry. The methods section has been updated to explicitly describe these methods.

2. The authors examine alpha4beta1 integrin, a major integrin regulating extravasation in T cells. In myeloid cells, however, alpha4beta1 is a relatively minor integrin while beta2 integrins (LFA1, Mac1) are more important. Does the MLCK210-containing pathway also regulate these beta2 integrins?

We have previously shown that integrin $\alpha4\beta1$ is indeed a major integrin in myeloid cells; it is expressed at very high levels in myeloid cells, where it regulates trafficking to tumors. Our articles in that have been published in *Cancer Research*, *Cancer Cell*, *PlosOne*, *Nature Communications*, *Cancer Immunology Research* and *Nature* demonstrate that only integrin $\alpha4\beta1$, but not other integrins including $\alpha M\beta2$ (CD11b), promotes adhesion of myeloid cells to endothelium and trafficking to tumors in vivo in response to tumor secreted factors. We have included additional data supporting the requirement for $\alpha4$ integrin in myeloid cell adhesion to endothelium (Figure 2a) and referred to our previously published results to place the current studies in the context of our extensive published work on this topic. Our studies also show that MLCK210 does not substantially regulate the activity of beta 2 integrins (Supplemental Figure 2a).

Does the pathway function in T cells? The authors use a number of different substrates to address this, but integrins are notoriously promiscuous in terms of ligand binding, meaning this is not sufficient.

MLCK210 deletion or inhibition actually promotes T cell recruitment to tumors, concomitant with decreased tumor size, indicating that MLCK210 is dispensable for T cell function. These data are now included in Figure 7 and 8.

Similarly, the authors do not show that $\alpha4\beta1$ integrin is being activated, since the only direct assay employed uses an activation epitope-specific anti-beta1 antibody, that will inform on the activation status of any beta1 containing integrins.

The ability of integrin $\alpha4\beta1$ to bind ligand, as shown in the VCAM-Fc binding studies, as well as ability to adhere to substrates, is a measure of the acquisition of activity or "activation".

3. Quantitative analysis of the integrin activity status (HUTS21 binding; VCAM-1 binding with suspension cells) should be performed instead of (or in addition) to the functional assay (cell adhesion) wherever a new player is being analyzed e.g. Figs 4C,G, 5F

We extensively show these data for MLCK210 and Myh9. We did not show these data for integrin $\alpha4\beta1$, Rap1 or PI3K γ because we have already published such studies.

4. The major assay employed in an analysis of cell adhesion under static conditions with cells adhering to VCAM-1 coated plastic or HUVECs. Here myeloid cells are calcein labelled and adhesion is quantified using a plate reader. A better way to analyze integrin function in the context of static adhesion would be an analysis of cell adhesion microscopically analyzing cell number and cell size (in pixels), to inform on adhesion as well as spreading, both of which require integrin activity.

Our adhesion assays were very brief, only 20 minutes after addition of cells. This assay has been extensively published by us over the past 15 years and was designed to avoid measuring adhesion strengthening or spreading. As integrin $\alpha4\beta1$ mediated activation has been shown to occur within nanoseconds after exposure to cytokine/chemokine in circulating cells, spreading is a secondary consequence of activation. We did evaluate cell spreading as a means to assess cell adhesion. Circulating myeloid cells have minimal cytoplasm, and cell spreading is difficult to detect. However, we now include a new Supplementary figure (Supplementary Figure 4) that examines $\alpha4\beta1$, paxillin, talin, myosin and actin localization in WT and MLCK210^{-/-} myeloid cells spreading on VCAM-1 coated coverslips. MLCK210^{-/-} cells remain round with no sign of cytoplasm spreading outward while WT cells extend a small amount of cytoplasm showing a slight but noticeable difference in shape.

5. Relatedly, the authors co-stimulate the cells with chemokine / cytokine to induce integrin activation. Stimuli employed are IL-1beta, IL-6, SDF-1alpha, tumour cell medium, TNF and VEGF-A. Intriguingly, all these stimuli induce a very similar degree of activation in the adhesion experiments, suggesting that the assay has not been carefully calibrated and the linear range of the assay has been exceeded.

Most importantly, it is unclear how some of the stimuli used can cause a significant activation, given that the cells analyzed are likely predominantly neutrophils, with a smaller degree of monocytes/macrophages. Circulating neutrophils do not express VEGF-R (although there has

been a description of a subtype of cancer neutrophils that do, Massena et al 2015 *Blood* 125, 2016), and neither do the vast majority of circulating monocytes (Nowall et al 2004 *Circulation* 110, 2699). Similarly, IL-1beta does not stimulate isolated neutrophils (Prince et al 2004 *Am J Pathol* 165, 1819). This demonstrates again the importance of carefully addressing point 1.

We have extensively published studies using these same cytokines to stimulate integrin $\alpha 4\beta 1$ activation on monocytes and granulocytes and to promote adhesion of myeloid cells to endothelium and VCAM-1 and we have previously titrated the effect of these factors and explored the mechanism of action¹⁻⁵. We have provided extensive description of our published results in the manuscript to place the current studies in the context of our extensive published work from the past several years, adding reference to our articles on this topic from *Cancer Research*, *Cancer Cell*, *PlosOne*, *Nature Communications*, *Cancer Immunology Research* and *Nature*.

¹Schmid, M.C., et al. Combined blockade of integrin- $\alpha 4\beta 1$ plus cytokines SDF-1 α or IL-1 β potently inhibits tumor inflammation and growth. *Cancer Res* **71**, 6965-75 (2011).

²Schmid, M.C., et al. Receptor tyrosine kinases and TLR/IL1Rs unexpectedly activate myeloid cell PI3ky, a single convergent point promoting tumor inflammation and progression. *Cancer Cell* **19**, 715-27 (2011).

³Schmid, M.C., et al. PI3-kinase γ promotes Rap1a-mediated activation of myeloid cell integrin $\alpha 4\beta 1$, leading to tumor inflammation and growth. *PLoS One* **8**, e60226 (2013).

⁴Kaneda, M.M., et al. PI3Ky is a molecular switch that controls immune suppression. *Nature* **539**, 437-42 (2016).

⁵Foubert P, Kaneda MM, Varner JA. PI3Ky Activates Integrin $\alpha 4$ and Promotes Immune Suppressive Myeloid Cell Polarization during Tumor Progression. *Cancer Immunol Res*. 2017 Nov;5(11):957-968

We do not agree with the reviewer that VEGF-A and IL1 β do not stimulate monocyte and granulocyte trafficking in vivo. While one can always find papers that differ in opinion, this issue has never come up on our extensive publication history on this topic. Many publications show critical roles of VEGF and IL1 in granulocyte and monocyte mediated trafficking. These articles include:

Avraham-Davidi I, Yona S, Grunewald M, et al. On-site education of VEGF-recruited monocytes improves their performance as angiogenic and arteriogenic accessory cells. *J Exp Med*. 2013;210(12):2611-2625.

Massena S, Christoffersson G, Vågesjö E, et al. Identification and characterization of VEGF-A-responsive neutrophils expressing CD49d, VEGFR1, and CXCR4 in mice and humans. *Blood*. 2015;126(17):2016-2026.

Christoffersson G, Vågesjö E, Vandooren J, Lidén M, Massena S, Reinert RB, Brissova M, Powers AC, Opendakker G, Phillipson M. VEGF-A recruits a proangiogenic MMP-9-delivering neutrophil subset that induces angiogenesis in transplanted hypoxic tissue. *Blood*. 2012 Nov 29;120(23):4653-62. doi: 10.1182/blood-2012-04-421040.

Chen, L.-C., Wang, L.-J., Tsang, N.-M., Ojcius, D.M., Chen, C.-C., OuYang, C.-N., Hsueh, C., Liang, Y., Chang, K.-P., Chen, C.-C. and Chang, Y.-S. (2012), Tumour inflammasome-derived IL-1 β recruits neutrophils and improves local recurrence-free survival in EBV-induced nasopharyngeal carcinoma. *EMBO Mol Med*, 4: 1276-1293.

Lloyd S. Miller, Ryan M. O'Connell, Miguel A. Gutierrez, Eric M. Pietras, Arash Shahangian, Catherine E. Gross, Ajaykumar Thirumala, Ambrose L. Cheung, Genhong Cheng, Robert L. Modlin, MyD88 Mediates Neutrophil Recruitment Initiated by IL-1R but Not TLR2 Activation in Immunity against *Staphylococcus aureus*, *Immunity*, Volume 24, Issue 1, 2006, 79-91.

Fasano MB, Cousart S, Neal S, McCall CE. Increased expression of the interleukin 1 receptor on blood neutrophils of humans with the sepsis syndrome. *J Clin Invest*. 1991;88(5):1452-1459.

Dmitrieva-Posocco O, Dzutsev A, Posocco DF, et al. Cell-Type-Specific Responses to Interleukin-1 Control Microbial Invasion and Tumor-Elicited Inflammation in Colorectal Cancer. *Immunity*. 2019;50(1):166-180.e7.

6. In addition to static adhesion assays, leukocyte adhesion should be analyzed under more physiological flow conditions in a parallel plate chamber.

To study the role of MLCK210 during adhesion under flow would add only modest new perspectives. Prior studies have extensively used flow chambers to model adhesion under flow and these methods documented the importance of integrin $\alpha 4\beta 1$ activation in adhesion of leukocytes under flow. We believe that our *in vivo* studies documenting the role of MLCK210 in the trafficking of myeloid cells into tumors (Figure 6a-e) and our prior studies documenting similar roles for $\alpha 4\beta 1$ and PI3K γ that were previously published in *Cancer Cell* provide more physiologically relevant evidence of the importance of the roles of these molecules under the shear stresses experienced by cells *in vivo*. We also do have access to a flow chamber under the current COVID crisis.

7. Figure 2F shows that MLCK210 knock-down cells have a migration defect. This is a very important observation, given that recruitment to tumors depends upon migration. Is this a transwell assay? The authors should perform time-lapse imaging on different substrates to identify whether the defect is due to an adhesion or a de-adhesion defect; fantastic reagents (e.g. function blocking antibodies) exist for human integrins that would allow the identification of the integrin required. Transendothelial migration should also be investigated.

We do not have the capability of providing time lapse imaging and cannot arrange to do so currently under COVID restrictions. The data on cell migration are not a major part of our paper and were only provided to show that there is some effect of MLCK inhibition on migration. Our *in vivo* studies documenting the role of MLCK210 in the trafficking of myeloid cells into tumors (Figure 6a-e) and our prior studies documenting similar roles for $\alpha 4\beta 1$ and PI3K γ that were previously published in *Cancer Cell* provide more physiologically relevant evidence of the importance of the roles of these molecules under the shear stresses experienced by cells *in vivo*.

8. To measure integrin clustering and colocalization of proteins involved in the pathway, the authors stimulate the cells with BSA- or CXCL12-coated beads. The beads are described as very large (9 μ m) and should be shown in the first instance used images – presumably the cells are undergoing frustrated phagocytosis?

We apologize for a major typographical error. The 9 μ m should have read as 0.9 μ m. Bead attachment sites are now highlighted with arrows.

The analysis of colocalization is not convincing. Rather than co-localizing the integrin appears to be localizing to the same area of the cell but further to the plasma membrane than MLCK210 (Fig 1e; S3A), Rap (Fig S3A) and Myosin (Fig 5e) giving the impression of sequential rather than co-

localized stain. At best there may be instances of partial co-localization. Use of a plasma membrane dye may be helpful here. Moreover, single representative cells are not sufficient to draw firm conclusions and analysis of a larger number of images should be presented. Please also note that integrin clustering is a molecular event that occurs on a scale too small to be visualized with an ordinary light microscope, so this term should be avoided, or else superresolution microscopy or immuno-EM should be employed to analyze molecular events.

We have used brighter images that highlight the regions of overlap of $\alpha 4\beta 1$ and MLCK staining (yellow) and annotated these images with arrows in Figure 1e. We also showed additional images in Supplementary Figures that show overlap (yellow) in obvious “clusters.” Figure 1g clearly shows that integrin $\alpha 4\beta 1$ in *Mlck* knockdown cells is not focally organized, while in control transfected cells, it is clearly organized into “clusters”.

9. Fig 6A,B show recruitment of adoptively transferred control or MLCK210-deficient myeloid cells to tumors or the spleen. It is unclear why the authors analyze recruitment to the spleen rather than the bone marrow and/or the lung where adoptively transferred cells tend to marginate; also blood cells should be shown. How many of the adoptively transferred cells do remain viable after 24 hours?

Recruitment to spleen is a passive event; this is a common homing site for leukocytes. This is a control to demonstrate the specificity of the MLCK210 homing pathway to the tumor. We have not determined the viability of adoptively transferred cells after 2 days but have previously shown that the cells that arrive in the tissues are intact and viable for up to 48 h.

10. The tumour data, especially those performed with bone marrow chimeras are very interesting, though do not appear overly connected to the work presented in the first 5 figures. Importantly the experiments presented do not show that tumour growth is dependent upon MLCK10 in myeloid cells – it could be due to any radiosensitive cell type(s).

To test the dependency on a particular immune cell type depletion experiments ought to be performed. In addition, experiments in which mice with established tumours are treated with the MW01 compound should be performed to test whether blocking MLCK210 is sufficient to reverse tumour growth.

We do connect the tumor data to the rest of the paper by showing first that short-term trafficking to tumors is inhibited in MLCK210^{-/-} cells similar to integrin activation and adhesion in endothelium. The observation that myeloid cell as well as macrophage accumulation in tumors is suppressed in MCLK210^{-/-} BMT and drug treated animals logically follows from these studies.

We previously published extensively (see references below) that integrin $\alpha 4\beta 1$ is required for myeloid cell trafficking to tumors and for tumor growth¹⁻². We furthermore showed that PI3K γ activates $\alpha 4\beta 1$ to promote trafficking to tumors and tumor growth². We then showed that Rap1 is required for PI3K γ mediated integrin activation, myeloid cell trafficking and tumor growth³. In this manuscript, we show that MLCK210 is required for PI3K γ dependent $\alpha 4\beta 1$ integrin activation and adhesion and for myeloid cell trafficking and tumor growth. Taken together, our new studies show that inhibition of PI3K γ , MLCK210, Myosin and Rap1 suppresses the same integrin $\alpha 4\beta 1$ activation pathway that promotes myeloid cell trafficking to tumors and results in tumor growth.

In each of these papers, we show that loss of Rap1 or MLCK210 in whole animal or bone marrow inhibits tumor growth. In Kaneda et al⁴ and Foubert et al⁵, we show that CD8⁺ T cell depletion

reversed the tumor growth inhibition conferred by $\alpha 4$ and PI3K γ mutations. Because we show that MCLK210 is a component of the same pathway, T cell depletion would very likely show reversal of the benefit provided by MLCK10 knockdown; such studies should be the focus of the next publication.

¹Schmid, M.C., et al. Combined blockade of integrin- $\alpha 4\beta 1$ plus cytokines SDF-1 α or IL-1 β potently inhibits tumor inflammation and growth. *Cancer Res* **71**, 6965-75 (2011).

²Schmid, M.C., et al. Receptor tyrosine kinases and TLR/IL1Rs unexpectedly activate myeloid cell PI3ky, a single convergent point promoting tumor inflammation and progression. *Cancer Cell* **19**, 715-27 (2011).

³Schmid, M.C., et al. PI3-kinase γ promotes Rap1a-mediated activation of myeloid cell integrin $\alpha 4\beta 1$, leading to tumor inflammation and growth. *PLoS One* **8**, e60226 (2013).

⁴Kaneda, M.M., et al. PI3Ky is a molecular switch that controls immune suppression. *Nature* **539**, 437-42 (2016).

⁵Foubert P, Kaneda MM, Varner JA. PI3Ky Activates Integrin $\alpha 4$ and Promotes Immune Suppressive Myeloid Cell Polarization during Tumor Progression. *Cancer Immunol Res.* 2017 Nov;5(11):957-968

Minor points.
This manuscript appears to have been put together rather hastily.
1. The methods are too brief for readers to repeat any of the experiments. Incubation/stimulation times, agonist concentrations, objectives used for microscopy details on image analysis etc are all missing.

Thank you for identifying this weakness. We have now added substantial detail to the methods section, as requested.

2. The figure legends do not put experiments into context and should be expanded. What is the migration assay shown in Fig 2F (this is also not mentioned in the methods section)? Definitions are also missing (e.g. TOPRO, Fig S3C). There are no details on statistical analysis performed, with any of in vitro data shown.

We have included more detail in the main text about the purpose of each study and more detail in figure legends. We also now include experimental details of the migration assays and the statistical analyses of all in vitro and all in vivo studies in the methods section.

3. It should be made clear throughout whether experiments were performed with human or mouse cells by clearly labelling the figures. Key experiments should be performed with both, since signaling is often not conserved between the two species.

We have now indicated on each figure which studies were performed with mouse and which with human cells. Where possible we have performed studies with both. For example, HUTS21 studies can only be performed with human cells, as the antibody is human specific. Similarly, we only performed studies with MLCK210 knockout cells using mouse cells.

4. Multiple Western blot-based experiments rely on a single representative example. Several experiments ought to be performed in each instance, signal quantification should be performed and shown. This is especially critical where Western blots are performed to inform on the activity status of a protein of interest.

Each of our studies, including Western blotting, have been performed repeatedly with similar results, as stated in our methods section. Original uncropped blots and original data underlying each graph are now provided with the manuscript. As our Western blots show exceptionally quantitatively clear results, we do not believe that signal quantification of gels will alter the result or the overall message of this paper.

All micrographs are missing scale bars.

These have now been added.

6. A section that informs on the statistical analysis of in vitro experiments is missing.

This has now been added.

7. Individual points ought to be presented rather than bar graphs, especially in Fig 6 but ideally throughout.

These have been added throughout the paper.

8. The timing of the data shown in Fig 6O,P should be indicated in the legend.

This has now been done.

Reviewers' Comments:

Reviewer #1:

Remarks to the Author:

The authors have addressed the previous comments in a satisfactory manner. I find merit in the effort made to incorporate and respond to all comments by reviewers. This has increased the quality of the manuscript.

Minor points:

- 1) I don't see the point of showing the cell counts per peak in each FACS profile for the ligand binding assays. My previous comments referred to the added value of normalizing VCAM1 binding to the total level of surface $\alpha 4\beta 1$ integrin, especially in Fig 3h where siRNA-mediated knockdown of Mlck is accompanied by a decrease in VCAM1 binding upon addition of Mn²⁺. It is intriguing that SDF-1 α induces a much greater increase in VCAM1 binding compared to Mn²⁺ in Fig 5e.
- 2) I am puzzled that there is a significant difference ($p=0.0001$) for the Mn²⁺ condition between WT and Mlck210^{-/-} samples in Fig 3g.
- 3) It may be difficult for the reader to fully appreciate the representation of statistical differences in the figures (e.g. Fig 2a, b, c, d, f, g; Fig 3a, g, h; Fig 4g) as they are too numerous and are not correctly aligned. It would be beneficial to use a simpler and more precise drawing.

Reviewer #2:

Remarks to the Author:

there have been some improvements from the previous version, especially in the data representation. However, I'm not completely satisfied to the answers of my concerns:

- 1) it is mentioned that the protein extracted from the band was sequenced and proteomic analysis performed, but no details on the procedure or raw data are provided. was an amino and c terminus sequencing? I can interfere that tandem mass spec might be used but not much more.
- 2) Thank you for providing the p value and the raw data. However, many typo have been done in the figures (possibly a wrong version). i.e. you should write $P<0.001$ when this is true, you should not omit the fourth digit when p value is 0.056 (fig5c for example), figure 3g there is absolutely no difference between MN²⁺ in the WT an in the Mlck210 but a $p=0.0001$ is assigned to this comparison. many other typo errors characterized almost all the figures. some $p=0.02$ became 0.002 or vice versa.
- 3) thank you for adding the full gel in supplementary figure 1. This partially address my concerns on the isoforms.
- 4) thank you for adding the IF analysis evaluating a little bit better the TME.
- 5) I appreciate that you included additional picture of colocalization. this helps a little bit however it whould have been better an automatic quantification of signal overlapping using a minimum of 50-100 cells. thee are many freeware programs or ImageJ app that can quantify the signal.

Additional concerns:

- 1) please double check the p value of the of the CD8 in figure 8G. The graph does not appear to be significant and the data provided in the zipfile for reviewers show that the T test p value of 0.076 that is obviously not significant. the relative statement of CD8 increase should be removed.
- 2) upon relooking at the figures I realize that Fig.1e SDF-1 α WT and fig.5i pFlag KD pictured exactly the same cell. I believe that this has been an oversight since both panel belong to the same experimental group. However it would be better to use a different picture.

Reviewer #3:

Remarks to the Author:

Having read this revised manuscript carefully I have come to realise that the cells that are referred to here as 'myeloid' here are monocytes/macrophages (CD11b positive mononuclear cells from

bone marrow). Since the cells that are reduced in the mouse cancer model are monocytes/macrophages, I would suggest the authors refer to monocytes/macrophages throughout the text including in the title rather than using the more misleading term 'myeloid cells' – since such a high percentage of bone marrow myeloid cells are neutrophils, and neutrophils are also the most abundant circulating leukocytes in human calling the monocytes/macrophages myeloid cells easily misleads the reader into thinking that this is an extremely heterogeneous mix of cells. This is what happened to me when I read the original manuscript.

Cells. The point-by-point states that the human myeloid cells were purified by flow cytometry to study the role of MLCK210 in granulocytes and monocytes. In contrast the methods section states that they were purified by gradient centrifugation and anti-CD11b affinity bead isolation and that populations were assessed by FACS analysis. Which is it? I would ask the authors to show the flow cytometrical analysis of these populations in the data supplement, and specify which markers were employed to ascertain purity of the cell types that were analysed.

The fact that the authors observe virtually identical adhesion with neutrophils in response to stimulation with an array of unusual stimuli remains puzzling as mentioned in my initial review. Unprimed, peripheral healthy donor blood derived neutrophils or those prepared from mouse bone marrow do not behave like TANs. They don't express VEGFR and they use CXCL12 for homing to the bone marrow of aged cells which upregulate CXCR4. The neutrophils ought to be analysed for activation markers (L-selectin shedding, surface Mac-1) to exclude they were primed during preparation. Neutrophils will spread considerably in a 10-20 minute adhesion assay. Therefore it makes sense to analyse both adhesion and spreading. The images provided in Fig S4 do not address this point.

T cells.

The fact that MLCK210-deficiency promotes CD8+ T cell recruitment to tumours does not indicate that MLCK210 is dispensable for T cell function. Rather it suggests that any function it may have is different to that observed with monocytes/macrophages. It will be easy to test whether T cells express MLCK210, and if they do, whether MLCK210-deficiency affects T cell alpha4 beta1. If it does not, this will strengthen this story significantly. As the authors will be aware, there are many T cell types outside of CD8+ cytotoxic T cells. Is recruitment of other T cell species also increased in MLCK210-deficiency? Parallel sections to the ones shown in Fig 7i should be used for additional staining for added insight into this question. Finally, to identify how large the impact of enhanced CD8+ T cell recruitment is on tumour development in the context of reduced monocyte/macrophage recruitment in MLCK210-deficiency the authors should perform antibody-mediated T cell depletion. The authors should also use clodronate-mediated depletion of monocytes/macrophages to confirm whether monocytes interfere with cytotoxic T cell recruitment into the tumour tissue.

Minor points

1. Fig 1A – silver-stained gels and legend are not lined up. The Gelsolin gel is pixelated.
2. VEGF is a growth factor; please correct (in text and figure legends)
3. Fig 2g shows transwell chemotaxis towards SDF-1alpha (concentration?) or tumour cell conditioned medium. According to the method this experiment analysed cell migration to integrin ligands (VCAM-1, vitronectin, fibronectin) on the underside of the transwell? Please show missing values.
4. Fig 2f,g the legend is grey but there are no grey bars in these graphs.
5. Integrin activation is quick – but nanoseconds (p8 line 7)? Please provide a primary reference or correct. Shamri et al Nat Immunol 2005 measured 0.4s for T cells LFA-1 activation.
6. Fig 6g, k; 7c – mean ± range should be shown rather than SEM.
7. The blots shown in Fig 4 a, d and f are not very clean. The data from the repeat experiments performed should be quantified and shown graphically.
8. Stats. There is no non-parametric Student's T test. Which test was used?

Summary of additions and modifications to the manuscript

All figures: All graphs have been simplified and p values now included in figure legends.

Figure 1 has been rearranged and updated with new analyses of integrin clustering.

1) Figure 1b: We have now added Tandem Mass Spectrometry profiles and the table of peptide data identifying integrin $\alpha4\beta1$ associated proteins MLCK210 and Gelsolin in Figure 1b. An image of the original silver stain gel can be found in Supplementary Figure 1.

2) Figure 1d: We have now added images and quantification of integrin $\alpha4\beta1$ clustering /aggregation in SDF-1 and BSA-stimulated WT and *Mlck210*^{-/-} cells in Figure 1d-e

3) Figure 1h: We now show select images and added quantification of integrin $\alpha4\beta1$ overlap with MLCK210 in *Mlck210*^{-/-} cells that were transduced to express WT or kinase dead (KD) MLCK210 in Figure 1h; additional images of MLCK210 or paxillin / $\alpha4$ co-localization are now found in Supplementary Figure 1c-f.

Figure 3: We now include in FACs profiles and quantification of integrin $\alpha4$ and integrin $\beta1$ single stains upon stimulation of cells in Supplementary Figure 4a-b. We also show that MLCK210 isoform is expressed in myeloid cells but not in T cells in Supplementary Figure 4c.

Figure 4: No change

Figure 5: MLCK210 and Paxillin immunofluorescence images were moved to Supplementary Figure 1d-e.

Figure 6: Facs profiles and quantification of myeloid cells in WT and *Mlck210*^{-/-} tumors have been added in Figure 6m-o.

Figure 7: No change

Figure 8: A new Figure 8 has been added that includes tumor studies in WT and *Mlck210*^{-/-} animals that were treated with either clodronate to reduce macrophage content or anti- CD8+ to deplete CD8+ T cells from tumors.

Figure 9: Former Figure 8 is now Figure 9.

Supplementary Figure 1: In Supplementary Figure 1a, we now show the exact silver-stained gel from which the integrin $\alpha4\beta1$ co-precipitating proteins MLCK210 and Gelsolin were isolated for Tandem Mass Spectrometry. In Supplementary Figure 1c-f, we now show images of integrin $\alpha4\beta1$ co-clustering with MLCK210 and paxillin upon SDF-1 stimulation. Supplementary Figure 1c shows images of multiple cells per field while Supplementary Figures 1d-f show images of single cells/field.

Supplementary Figure 2: No change

Supplementary Figure 3: 3a- We show Facs profiles and expression levels of integrin $\alpha4$ and $\beta1$ cell surface expression levels in myeloid cells that had been stimulated with cytokines or

BSA. We also show a Western blot documenting the expression of MLCK210 in myeloid cells but not T cells.

Supplementary Figure 4: We moved MLCK210 and Paxillin images to Supplementary Figure 1.

Supplementary Figure 5: This figure illustrates the FACS gating schemes used in this manuscript.

Detailed responses to Reviewers

Reviewer #1 (Remarks to the Author):

The authors have addressed the previous comments in a satisfactory manner. I find merit in the effort made to incorporate and respond to all comments by reviewers. This has increased the quality of the manuscript.

Thank you for your positive comments.

Minor points:

1) I don't see the point of showing the cell counts per peak in each FACS profile for the ligand binding assays. My previous comments referred to the added value of normalizing VCAM1 binding to the total level of surface $\alpha 4\beta 1$ integrin, especially in Fig 3h where siRNA-mediated knockdown of Mlck is accompanied by a decrease in VCAM1 binding upon addition of Mn²⁺. It is intriguing that SDF-1 α induces a much greater increase in VCAM1 binding compared to Mn²⁺ in Fig 5e.

Thank you for these comments. We now include data in Supplementary Figure 4a-b evaluating the effect of MLCK210 inhibition on integrin $\alpha 4$ and $\beta 1$ expression levels. These data show that MLCK210 inhibition/deletion has no effect on integrin $\alpha 4$ or $\beta 1$ expression levels, thus allowing us to conclude that changes in VCAM1 binding upon MLCK210 inhibition results from change in integrin activation rather than changes in cell surface expression levels.

2) I am puzzled that there is a significant difference ($p=0.0001$) for the Mn²⁺ condition between WT and Mlck210^{-/-} samples in Fig 3g.

Thank you for noting this. This was a typographical error that has been corrected. There is no significant difference.

3) It may be difficult for the reader to fully appreciate the representation of statistical differences in the figures (e.g. Fig 2a, b, c, d, f, g; Fig 3a, g, h; Fig 4g) as they are too numerous and are not correctly aligned. It would be beneficial to use a simpler and more precise drawing.

Thank you for the suggestion. All graphs have been simplified and p values are now included in figure legends when they were not included in the figures.

Reviewer #2 (Remarks to the Author):

there have been some improvements from the previous version, especially in the data

representation. However, I'm not completely satisfied to the answers of my concerns:
1) it is mentioned that the protein extracted from the band was sequenced and proteomic analysis performed, but no details on the procedure or raw data are provided. was an amino and c terminus sequencing? I can interfere that tandem mass spec might be used but not much more.

We have now included the actual tandem mass spec data in Figure 1b and the image of the original silver-stained gel from which protein were extracted in Supplementary Figure 1a.

2) Thank you for providing the p value and the raw data. However, many typo have been done in the figures (possibly a wrong version). i.e. you should write $P < 0.001$ when this is true, you should not omit the fourth digit when p value is 0.056 (fig5c for example), figure 3g there is absolutely no difference between MN2+ in the WT an in the *Mlck210* but a $p = 0.0001$ is assigned to this comparison. many other typo errors characterized almost all the figures. some $p = 0.02$ became 0.002 or vice versa.

We apologize for the many errors and the rounding off of the p values. After conferring with the editor, the figures have been simplified and now the full p values including all 4 digits have been included in figure legends and in the source data. The error identified in Figure 3g has been corrected.

3) thank you for adding the full gel in supplementary figure 1. This partially address my concerns on the isoforms.

We have now included the actual tandem mass spec data in Figure 1b and image of the original silver-stained gel in Supplementary Figure 1a.

4) thank you for adding the IF analysis evaluating a little bit better the TME.

Thank you

5) I appreciate that you included additional picture of colocalization. this helps a little bit however it would have been better an automatic quantification of signal overlapping using a minimum of 50-100 cells. there are many freeware programs or ImageJ app that can quantify the signal.

We have now used Image J to quantify integrin clustering in 70 to 125 WT and *Mlck210*^{-/-} myeloid cells in Figure 1d and to quantify the overlap of MLCK210 and integrin $\alpha 4\beta 1$ in 40-140 cells per condition in Figure 1h.

Additional concerns:

1) please double check the p value of the of the CD8 in figure 8G. The graph does not appear to be significant and the data provided in the zipfile for reviewers show that the T test p value of 0.076 that is obviously not significant. the relative statement of CD8 increase should be removed.

We used Mann Whitney nonparametric significance test as the data were not normally distributed. By Mann Whitney test, the p value is 0.0244.

2) upon relooking at the figures, I realize that Fig.1e SDF-1a WT and fig.5i pFlag KD pictured exactly the same cell. I believe that this has been an oversight since both panels belong to the

same experimental group. However, it would be better to use a different picture.

Thank you for this helpful comment. We now provide unique images for each condition for these images, both of which have been moved to Supplementary Figure 1.

Reviewer #3 (Remarks to the Author):

Having read this revised manuscript carefully I have come to realize that the cells that are referred to here as 'myeloid' here are monocytes/macrophages (CD11b positive mononuclear cells from bone marrow). Since the cells that are reduced in the mouse cancer model are monocytes/macrophages, I would suggest the authors refer to monocytes/macrophages throughout the text including in the title rather than using the more misleading term 'myeloid cells' – since such a high percentage of bone marrow myeloid cells are neutrophils, and neutrophils are also the most abundant circulating leukocytes in human calling the monocytes/macrophages myeloid cells easily misleads the reader into thinking that this is an extremely heterogeneous mix of cells. This is what happened to me when I read the original manuscript.

Our data in Figure 2c demonstrate roles for MLCK210 in mediating integrin $\alpha 4\beta 1$ activation in both monocytes and granulocytes. We have now added data in Figure 6m-n demonstrating that MLCK210 inhibition reduces both monocyte/macrophage and granulocyte accumulation in tumors. Our data in Figure 9f (formerly Figure 8f) also shows that the MLCK210 inhibitor MW01-022AZ suppresses both monocyte and granulocyte accumulation in tumors. Although our BMT data in Figure 7 did not show a reduction in granulocyte accumulation, the preponderance of evidence indicates that MLCK210 plays a role in the recruitment of both types of myeloid cells.

Cells. The point-by-point states that the human myeloid cells were purified by flow cytometry to study the role of MLCK210 in granulocytes and monocytes. In contrast the methods section states that they were purified by gradient centrifugation and anti-CD11b affinity bead isolation and that populations were assessed by FACS analysis. Which is it? I would ask the authors to show the flow cytometrical analysis of these populations in the data supplement, and specify which markers were employed to ascertain purity of the cell types that were analyzed.

We apologize for any confusion. We have clarified the methods section describing these approaches. We routinely isolate BM from mice, perform red blood cell lysis and then histopaque gradient centrifugation. This yields mononuclear cells including monocytes and lymphocytes. To purify CD11b+ cells, we then further isolated cells by CD11b+ magnetic bead purification. To separately purify granulocytes and monocytes for figure 2c, we also then performed flow cytometry to sort for CD11b+Gr1lo monocytes and CD11b+Gr1hi granulocytes. These methods are clearly delineated in the methods sections. For human myeloid cells, we isolated CD11b+ cells from plasmapheresis samples by gradient centrifugation and anti-CD11b affinity bead isolation.

The fact that the authors observe virtually identical adhesion with neutrophils in response to stimulation with an array of unusual stimuli remains puzzling as mentioned in my initial review. Unprimed, peripheral healthy donor blood derived neutrophils or those prepared from mouse bone marrow do not behave like TANs. They don't express VEGFR and they use CXCL12 for homing to the bone marrow of aged cells which upregulate CXCR4. The neutrophils ought to be analyzed for activation markers (L-selectin shedding, surface Mac-1) to exclude they were

primed during preparation. Neutrophils will spread considerably in a 10-20 minute adhesion assay. Therefore, it makes sense to analyze both adhesion and spreading. The images provided in Fig S4 do not address this point.

We previously demonstrated in Schmid et al *Cancer Research* and *Cancer Cell* 2011 (see reference below) that integrin $\alpha 4\beta 1$ mediates adhesion and trafficking of both monocytes and neutrophils in response to these cytokines and chemokines. The data we present in Figure 2c demonstrates that granulocytes minimally adhere to endothelium or VCAM-1 without stimulation. Upon stimulation, they do adhere in an integrin $\alpha 4\beta 1$ -dependent manner. Importantly, Thalia Papayannopoulou's lab showed that integrin $\alpha 4\beta 1$ plays a key role in the recruitment of neutrophils to thioglycolate stimulated peritoneum in WT but not in integrin $\alpha 4$ deficient animals (see reference below). Since we have extensively published in leading journals that neutrophils adhere to endothelium and VCAM in response to stimulation by these same cytokines and our data and the literature clearly demonstrate this property, we must respectfully disagree in this instance. Since cell migration is a secondary event that is dependent on adhesion and spreading, our analysis of cell migration serves as a surrogate assay for cell spreading assays.

Ulyanova T, Priestley GV, Banerjee ER, Papayannopoulou T. Unique and redundant roles of alpha4 and beta2 integrins in kinetics of recruitment of lymphoid vs myeloid cell subsets to the inflamed peritoneum revealed by studies of genetically deficient mice. *Exp Hematol.* 2007;35(8):1256-1265. doi:10.1016/j.exphem.2007.04.015

Schmid MC, Avraamides CJ, Foubert P, et al. Combined blockade of integrin- $\alpha 4\beta 1$ plus cytokines SDF-1 α or IL-1 β potently inhibits tumor inflammation and growth. *Cancer Res.* 2011;71(22):6965-6975. doi:10.1158/0008-5472.CAN-11-0588

Schmid MC, Avraamides CJ, Dippold HC, et al. Receptor tyrosine kinases and TLR/IL1Rs unexpectedly activate myeloid cell PI3ky, a single convergent point promoting tumor inflammation and progression. *Cancer Cell.* 2011;19(6):715-727. doi:10.1016/j.ccr.2011.04.016

T cells.

The fact that MLCK210-deficiency promotes CD8+ T cell recruitment to tumours does not indicate that MLCK210 is dispensable for T cell function. Rather it suggests that any function it may have is different to that observed with monocytes/macrophages. It will be easy to test whether T cells express MLCK210, and if they do, whether MLCK210-deficiency affects T cell alpha4 beta1. If it does not, this will strengthen this story significantly. As the authors will be aware, there are many T cell types outside of CD8+ cytotoxic T cells. Is recruitment of other T cell species also increased in MLCK210-deficiency? Parallel sections to the ones shown in Fig 7i should be used for additional staining for added insight into this question. Finally, to identify how large the impact of enhanced CD8+ T cell recruitment is on tumour development in the context of reduced monocyte/macrophage recruitment in MLCK210-deficiency the authors should perform antibody-mediated T cell depletion. The authors should also use clodronate-mediated depletion of monocytes/macrophages to confirm whether monocytes interfere with cytotoxic T cell recruitment into the tumour tissue.

We have now included new data in Supplementary Figure 7 that demonstrates that CD11b+ cells express MLCK210 while T cells do not.

We also added new data in Figure 8 g-j that demonstrate that tumors implanted in the *Mlck210*^{-/-} background recruit substantially more CD8⁺ T cells (approximately 5 times more CD8⁺ T cells) to tumors, while CD4⁺ T cells content is not significantly different in tumors in *Mlck210*^{-/-} and WT mice. We also demonstrate in Figure 8g-l that antibody mediated depletion of CD8⁺ T cells reverses the beneficial effect of *Mlck210* deletion on tumor growth and provide data from tumor and spleen that CD8 cells were substantially depleted from antibody treated animals.

As requested, we also evaluated the effect of clodronate on tumor growth in the WT and *Mlck210*^{-/-} background in new Figure 8a-f. Clodronate reduced tumor growth in the WT background to the same degree as *Mlck210* deletion in *Mlck210*^{-/-} mice, it had no additive effect on tumor growth in the *Mlck210*^{-/-} background, even though macrophages were similarly depleted in both WT and *Mlck210* knockout mice (Figure 8d-f). These results are similar to those we observed and published in Kaneda, M.M., et al. PI3K γ is a molecular switch that controls immune suppression. *Nature* **539**, 437-42 (2016). In these studies, we found that clodronate mediated depletion of monocyte/macrophages did not synergize with PI3K γ inhibition, as both approaches similarly depleted monocyte/macrophages from tumors. We concluded that synergy only occurs when distinct biological targets or mechanisms are inhibited and that clodronate and PI3K γ or MLCK210 deletion or inhibition both prevent the accumulation of tumor promoting monocyte/macrophages.

Minor points

1. Fig 1A – silver-stained gels and legend are not lined up. The Gelsolin gel is pixelated.

We have revised Figure 1 to include additional Tandem Mass Spectrometry data. We have aligned the silver-stained gels and legend. We have improved the resolution of the silver stained gels to the maximum extent possible.

2. VEGF is a growth factor; please correct (in text and figure legends)

We had previously checked on the use of this phrase and found that the definition of cytokine in the Oxford dictionary: “any of a number of substances, such as interferon, interleukin, and growth factors, which are secreted by certain cells of the immune system and have an effect on other cells”. If the copy editors prefer us to use growth factor, we will be happy to change this.

3. Fig 2g shows transwell chemotaxis towards SDF-1 α (concentration?) or tumour cell conditioned medium. According to the method this experiment analysed cell migration to integrin ligands (VCAM-1, vitronectin, fibronectin) on the underside of the transwell? Please show missing values.

Migration assays in methods have been updated. The assay in Fig 2g represents chemotaxis (no coating with ECM) stimulation by SDF-1 α or TCM. The concentration of SDF-1 α is now given. All p values now present. No other values are missing.

4. Fig 2f,g the legend is grey but there are no grey bars in these graphs.

Thank you for noting this. The colors have been corrected.

5. Integrin activation is quick – but nanoseconds (p8 line 7)? Please provide a primary reference or correct. Shamri et al *Nat Immunol* 2005 measured 0.4s for T cells LFA-1 activation.

We apologize for the mistake. It is 0.1s in myeloid cells, corrected in text, reference by Grabovsky et al is now included.

6. Fig 6g, k; 7c – mean \pm range should be shown rather than SEM.

A graph demonstrating mean \pm range is included now in the source data file for these figure subsections.

7. The blots shown in Fig 4 a, d and f are not very clean. The data from the repeat experiments performed should be quantified and shown graphically.

We have now incorporated cleaner blots for these figures.

8. Stats. There is no non-parametric Student's T test. Which test was used?

All significance testing was performed in Prism by Graphpad software by unpaired two-sample Student's *t* test for two groups with normally distributed data and by Mann Whitney rank test for two groups when data were not normally distributed. Multiple group analyses were analyzed by one-way Anova with Tukey's posthoc testing for multiple pairwise testing (multiple groups) when data were normally distributed and a Wilcoxon rank sum test (two groups) when data were not.

Reviewers' Comments:

Reviewer #1:

Remarks to the Author:

The authors here provide additional results in response to reviewer comments that support and reinforce the findings of this study. However, I am only moderately satisfied with their response to my invitation to measure $\alpha 4\beta 1$ integrin expression to normalize VCAM1 ligand binding measurements. In Supplemental Figure 4, the authors only measure surface expression of integrin $\alpha 4$ and not $\beta 1$, contrary to what they state in the text (lines 215-216): "Depletion of MLCK210 had no impact on integrin $\alpha 4$ or $\beta 1$ cell surface expression levels (Supplementary Figure 4a-b)". I suggest confirming the $\beta 1$ integrin expression levels to maintain their conclusion. Additionally, $\alpha 4\beta 1$ integrin expression is not tested after treatment with anti-Mlck siRNA with respect to Figure 3h where siRNA-mediated Mlck inactivation is accompanied by a decrease in binding to VCAM1 upon addition of Mn²⁺, which is intriguingly not found in MLCK210^{-/-} cells in Figure 3g.

Reviewer #2:

Remarks to the Author:

The authors have satisfactorily addressed my previous concerns

Reviewer #3:

Remarks to the Author:

This manuscript by Schmid et al has been dramatically improved by this revision. It is particularly strengthened by the new data presented in Fig 8. The majority of my concerns have been addressed.

I have some minor remaining concerns:

- Fig S2B+C show the same cell twice (pFlag WT + BSA)
- some of the cells in Fig S2C+E and S5C (pFlag WT + SDF1 α) are very pixelated suggesting they were taken at a different resolution or processed in a different fashion
- Western blots (e.g. Fig 4) - only representative blots are shown. While these are now cleaner than in the previous version, I would welcome densitometry of repeat experiments in addition to the representative examples (esp for pMLC in panel F). At present there is not even an indication of how many repeats were performed.
- Fig 6 has no panel O (which is called out in the text)
- Fig 7B is this mRNA expression (as per ms text) or genomic DNA (as per figure legend)?

Response to Reviewers

Reviewer 1:

The authors here provide additional results in response to reviewer comments that support and reinforce the findings of this study. However, I am only moderately satisfied with their response to my invitation to measure $\alpha 4\beta 1$ integrin expression to normalize VCAM1 ligand binding measurements. In Supplemental Figure 4, the authors only measure surface expression of integrin $\alpha 4$ and not $\beta 1$, contrary to what they state in the text (lines 215-216): "Depletion of MLCK210 had no impact on integrin $\alpha 4$ or $\beta 1$ cell surface expression levels (Supplementary Figure 4a-b)". I suggest confirming the $\beta 1$ integrin expression levels to maintain their conclusion. Additionally, $\alpha 4\beta 1$ integrin expression is not tested after treatment with anti-Mlck siRNA with respect to Figure 3h where siRNA-mediated Mlck inactivation is accompanied by a decrease in binding to VCAM1 upon addition of Mn²⁺, which is intriguingly not found in MLCK210^{-/-} cells in Figure 3g.

Response: We measured integrin beta 1 cell surface expression levels and found there is a slight, but significant decrease in beta 1 expression levels in *Mlck210*^{-/-} cells. However, integrin $\alpha 4\beta 1$ surface expression is entirely dependent upon the level of the alpha chain, since the beta 1 chain can be utilized by over 10 different alpha chains. We have added this data to Supplementary Figure 4 with discussion of this slight decrease. We decided not to knockdown MLCK and perform this study since the only proven siRNAs knockdown all isoforms of MLCK, and we would be unable to examine selective effects of MLCK210 knockdown. We can speculate that the data in Figure 3g suggest that other isoforms of MLCK such as MLCK108 may also have an impact on integrin expression or function. We believe further investigation into this effect would distract from the goal of this paper and will not benefit the paper. As Reviewer 2 had questioned the need for siRNA studies at all, we could easily remove these data from the paper if required.

Reviewer #3

I have some minor remaining concerns:

- Fig S2B+C show the same cell twice (pFlag WT + BSA)

The reviewer is right. This was an oversight and we replaced the images in Figure S2C to ensure that all are unique images.

Some of the cells in Fig S2C+E and S5C (pFlag WT + SDF1alpha) are very pixellated suggesting they were taken at a different resolution or processed in a different fashion.

This was the result of inattention to image resolution at the time of making the adobe illustrator figures and we have corrected this.

Western blots (e.g. Fig 4) - only representative blots are shown. While these are now cleaner than in the previous version, I would welcome densitometry of repeat experiments in addition to the representative examples (esp for pMLC in panel F). At present there is not even an indication of how many repeats were performed.

We provided densitometry for panel f. These blots were performed twice. Regrettably, the MLC antibodies were not very clean and the panel shown is the best we could do.

- Fig 6 has no panel O (which is called out in the text)

This has been corrected.

-Fig 7B is this mRNA expression (as per ms text) or genomic DNA (as per figure legend)?

This was genotyping from mouse leukocytes, so it was genomic PCR. This has been corrected.